# SpatialReward: Bridging the Perception Gap in Online RL for Image Editing via Explicit Spatial Reasoning

Yancheng Long [*1]  Yankai Yang [*1]  Hongyang Wei [2]  Wei Chen [3]  Tianke Zhang [4]  Haonan Fan [4]  Changyi Liu [4]
Kaiyu Jiang [4]  Jiankang Chen [4]  Kaiyu Tang [4]  Bin Wen [4]  Fan Yang [4]  Tingting Gao [4]  Han Li [4]  Shuo Yang [1]

https://lorangan-ddup.github.io/SpatialReward/

## Abstract

Online Reinforcement Learning (RL) offers a promising avenue for complex image editing but is currently constrained by the scarcity of reliable and fine-grained reward signals. Existing evaluators frequently struggle with a critical perception gap we term "Attention Collapse," where models neglect cross-image comparisons and fail to capture fine-grained details, resulting in inaccurate perception and miscalibrated scores. To address these limitations, we propose **Spatial-Reward**, a reward model that enforces precise verification via explicit spatial reasoning. By anchoring reasoning to predicted edit regions, SpatialReward grounds semantic judgments in pixel-level evidence, significantly enhancing evaluative accuracy. Trained on a curated 260k spatial-aware dataset, our model achieves state-of-the-art performance on MMRB2 and EditReward-Bench, and outperforms proprietary evaluators on our proposed **MultiEditReward-Bench**. Furthermore, SpatialReward serves as a robust signal in online RL, boosting OmniGen2 by +0.90 on GEdit-Bench—surpassing the leading discriminative model and doubling the gain of GPT-4.1 (+0.45). These results demonstrate that spatial reasoning is essential for unlocking effective alignment in image editing.

## 1. Introduction

Instruction-guided image editing (Brooks et al., 2023; Zhang et al., 2023; Xu et al., 2025) has advanced rapidly, moving

---
*Equal contribution . Bin Wen is the project leader. [1]Harbin Institute of Technology, Shenzhen [2]Tsinghua Shenzhen International Graduate School, Tsinghua University [3]The Hong Kong University of Science and Technology [4]Kuaishou Technology. Correspondence to: Shuo Yang <shuoyang@hit.edu.cn>, Bin Wen <wenbin@kuaishou.com>.

*Proceedings of the 43rd International Conference on Machine Learning*, Seoul, South Korea. PMLR 306, 2026. Copyright 2026 by the author(s).

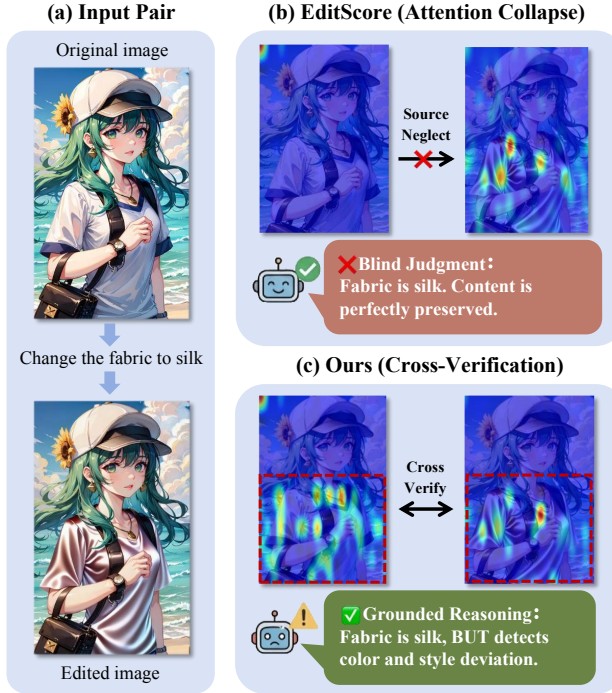

*Figure 1.* **Visualizing the Cross-Image Attention Gap. (a) Input Pair:** An editing instruction ("Change the fabric to silk") is executed, but with subtle inconsistencies. **(b) Baseline (Attention Collapse):** Due to source neglect, the baseline fails to attend to the reference image, leading to a blind judgment that incorrectly approves the edit. **(c) SpatialReward (Cross-Verification):** By anchoring reasoning to explicit spatial regions (red boxes), our model restores cross-image attention, enabling grounded verification that correctly detects the style deviation.

from simple style transfer to the precise editing of complex scenes. These tasks demand the reliable execution of multiple instructions while preserving non-target regions. However, current models often face a dilemma where they successfully execute the edit but inadvertently compromise the source identity or consistency, such as altering the original structure or background style. This issue exposes the limitations of pure Supervised Fine-Tuning (SFT), which tends to fit the data average and struggles with long-tail or compositional cases.

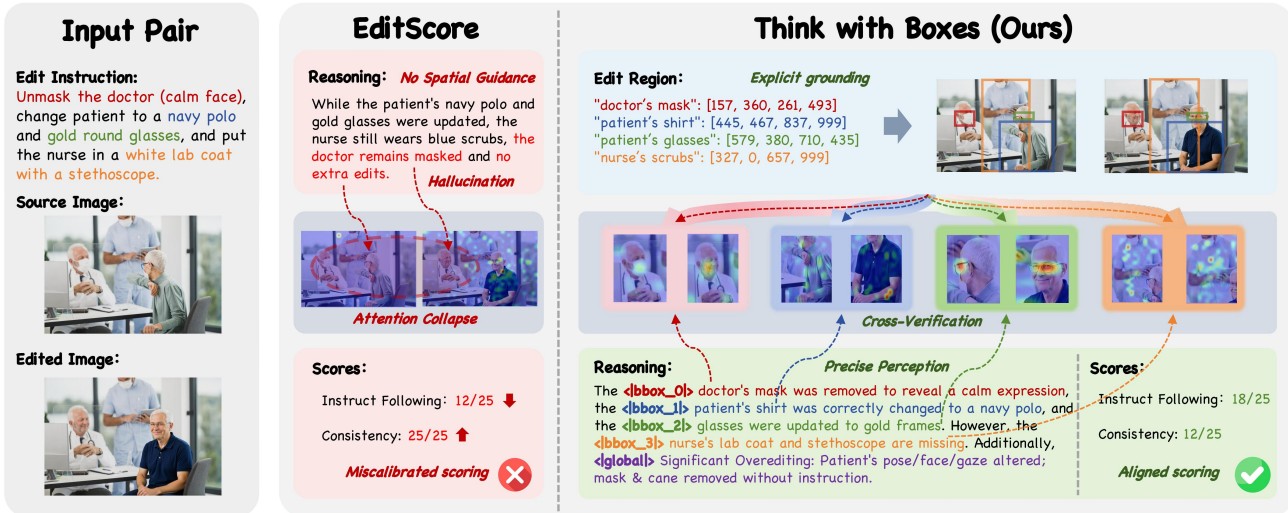

*Figure 2.* **Overview of SpatialReward and Comparison with Baseline.** (Left) The baseline (EditScore) lacks spatial guidance, leading to Attention Collapse and hallucinatory judgments; specifically, it overlooks the removal of the doctor's mask and the alteration of the patient's pose. (Right) Our SpatialReward employs a Think-with-Boxes mechanism: it first predicts bounding boxes (Edit Region) and injects them as interleaved tokens to anchor the subsequent reasoning. This enforces cross-verification (visualized by rectified attention maps), enabling precise detection of fine-grained inconsistencies (e.g., missing mask, altered pose) and ensuring aligned scoring.

In contrast, Online Reinforcement Learning (Online RL) treats editing as an interactive trial-and-error process, allowing the policy to explore out-of-distribution data and align with human preference. This paradigm, successfully demonstrated in LLMs (Ouyang et al., 2022; Touvron et al., 2023; Guo et al., 2025), has recently been advanced in generative models by methods like Flow-GRPO (Liu et al., 2025a) and Dance-GRPO (Xue et al., 2025). However, the effectiveness of these powerful optimizers relies heavily on the availability of a reward model that is reliable, efficient, interpretable, and spatially aware at a fine granularity.

Current reward mechanisms reveal three critical limitations when applied to interactive Online RL training for image editing.

First, pairwise rewards focus on relative ranking. While benchmarks like MMRB2 (Hu et al., 2025) show strong zero-shot performance for closed-source models, this relative comparison fails to provide the absolute scalar signals crucial for Online RL. Converting rankings introduces ambiguity, and the required pairwise inference imposes a prohibitive computational burden (often $O(N^2)$), creating unacceptable latency for online optimization.

Second, pointwise discriminative models, such as EditReward (Wu et al., 2025d), train a linear head atop VLM embeddings to regress preference scores. These models lack an explicit reasoning path and rely on costly human labels, limiting their scalability.

Finally, pointwise generative "MLLM-as-a-judge" methods offer a promising direction by explicitly modeling reason-

ing chains. However, they struggle with the unique demand of image editing: rigorous *cross-image region comparison*. Lacking explicit spatial guidance to anchor this comparison, even advanced models like GPT-5 suffer from a fundamental perception gap: they struggle to align and verify fine-grained details across images. This deficiency propagates and intensifies during distillation, specifically in bespoke reward models like EditScore (Luo et al., 2025). As visualized in Fig. 1(b), it manifests as "Attention Collapse"—where the model's focus, instead of attending to the source context, collapses into a blind sink state. This effectively renders the source image invisible and causes the critical task of cross-image comparison to degenerate into single-image evaluation, where inconsistencies with the source context are easily overlooked, leading to inaccurate scoring that diverges significantly from human preference.

To bridge this perception gap, we argue that reliable reward modeling must be built upon **explicit spatial reasoning**, which anchors perception to ensure precise verification and accurate scoring. To this end, we introduce **SpatialReward**, the first framework to integrate explicit spatial reasoning into generative pointwise evaluation for image editing. Our core "Think-with-Boxes" mechanism (Fig. 2) breaks the attention collapse by predicting edit-relevant spatial coordinates and utilizing interleaved tokens to anchor textual reasoning. This strategy compels the model to perform pixel-level verification between corresponding regions in the original and edited images, thereby ensuring that the final scalar reward faithfully reflects fine-grained edit quality. As visualized in Fig. 1(c), this rectified attention distribution enables the precise detection of subtle inconsistencies—such as un-

intended design changes—that were previously ignored by implicit baselines.

To support the framework, we construct SPATIALREWARD-260K. We impose spatial priors on the reasoning of the teacher model via explicit spatial coordinates to distill high-quality region-level reasoning traces. We then train in stages, moving from SFT to GRPO, to strengthen spatial reasoning and enforce scoring consistency.

We evaluate SpatialReward on three image-editing reward benchmarks: MMRB2 (Hu et al., 2025), EditReward-Bench (Luo et al., 2025), and our new MultiEditReward-Bench (MER-Bench). SpatialReward (8B) demonstrates superior performance across all metrics. Specifically, it improves over the generative baseline EditScore-8B by **+11.3%** on EditReward-Bench and **+9.1%** on MMRB2, surpassing the leading discriminative evaluator EditReward and all advanced proprietary models. Furthermore, it shows significant practical value in Online RL, lifting OmniGen2's (Wu et al., 2025b) performance on GEdit-Bench (Liu et al., 2025b) by +0.90, a margin nearly double that of GPT-4.1 (+0.45). These results indicate that fine-grained feedback with explicit spatial awareness is key to efficiently enhancing the efficacy of Online RL for image editing.

Our main contributions are summarized as follows:

- We identify the Perception Gap in MLLM-based evaluators, finding that the lack of spatial anchors leads to Attention Collapse, and demonstrate that explicit spatial grounding is essential to bridge this gap.

- We propose SpatialReward, the first framework to integrate explicit spatial reasoning into generative pointwise evaluation for image editing. To facilitate this, we introduce SPATIALREWARD-260K, a large-scale dataset containing high-quality spatial reasoning traces.

- We release MultiEditReward-Bench (MER-Bench), a benchmark constructing complex multi-region compositions to rigorously challenge the spatial perception and verification capabilities of reward models.

- Extensive experiments demonstrate that SpatialReward achieves state-of-the-art results on public benchmarks and significantly enhances downstream editing performance via Online RL, surpassing proprietary judges.

## 2. Related Work

### 2.1. Instruction-Guided Image Editing and Alignment

Early instruction-following editing models relied primarily on Supervised Fine-Tuning (SFT) over synthetic datasets.

Pioneers like InstructPix2Pix (Brooks et al., 2023) and MagicBrush (Zhang et al., 2023) demonstrated the efficacy of training diffusion models on paired data. Related generation and editing tasks also cover stylized handwritten text (Wang et al., 2026). Recent advances have integrated Multimodal LLMs (MLLMs) to enhance instruction understanding, as seen in MGIE (Fu et al., 2023) and OmniGen (Wu et al., 2025b). While effective, SFT-based methods tend to collapse to the mode of training data and often struggle with complex, compositional instructions. To address this, Reinforcement Learning (RL) has been introduced to align generative models with human preferences. Seminal works in text-to-image synthesis, such as ImageReward (Xu et al., 2023) and DPOK (Fan et al., 2023), utilized RLHF to improve aesthetic quality. More recently, algorithms like DDPO (Black et al., 2023) and D3PO (Yang et al., 2024) have enabled direct optimization of diffusion models. This paradigm is now extending to editing, with methods like Flow-GRPO (Liu et al., 2025a) and Dance-GRPO (Xue et al., 2025) leveraging stochastic exploration to escape local optima. However, the efficacy of these alignment algorithms is fundamentally limited by the quality of the feedback signal; without a reliable, fine-grained reward model, even powerful optimizers are prone to reward hacking or suboptimal convergence.

### 2.2. Reward Modeling for Generation and Editing

Standard T2I metrics like CLIP-Score (Radford et al., 2021) and PickScore (Kirstain et al., 2023) evaluate holistic alignment but lack granularity. Recent works address this by decomposing evaluation into multi-dimensional sub-criteria (e.g., aesthetics, semantics) derived from human preferences, as seen in MPS (Zhang et al., 2024), VisionReward (Xu et al., 2024), and HPSv3 (Ma et al., 2025). In the editing domain, methods additionally require cross-image verification. EditReward (Wu et al., 2025d) trains a discriminative regression head, while a concurrent work (Yang et al., 2026) adds auxiliary language supervision. To leverage the reasoning of strong models, UniPic 2.0 (Wei et al., 2025) uses GPT-4.1 via VIEScore (Ku et al., 2024), and EditScore (Luo et al., 2025) distills this capability. Another emerging paradigm, adopted by RewardDance (Wu et al., 2025c) and OneReward (Gong et al., 2025), derives scalar rewards directly from the token probabilities of generative "Yes/No" responses across these dimensions. However, despite these advances, most methods rely on implicit feature matching without explicit spatial grounding, leading to attention collapse and unreliable evaluation in complex editing scenarios.

### 2.3. Visual Reasoning and Spatial Grounding

Vision-language models (VLMs) like Shikra (Chen et al., 2023), Qwen-VL (Bai et al., 2023), Kosmos-2 (Peng et al., 2023), and Ferret (You et al., 2023) have demonstrated that

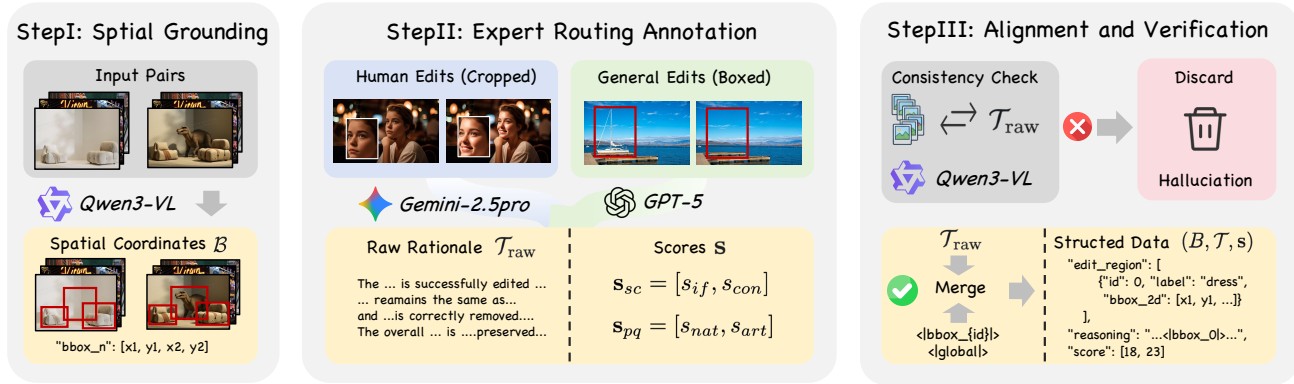

*Figure 3.* **Illustration of the Spatial-Prior-Guided Data Pipeline.** We construct a highly structured dataset by leveraging spatial priors. This involves spatial grounding via Qwen-3-VL, expert routing for reasoning annotations (using Gemini and GPT series), and a strict alignment verification process.

predicting explicit spatial coordinates strengthens object-attribute binding. While Chain-of-Thought (CoT) reasoning has reduced hallucinations in VQA, current image editing reward models have yet to leverage these advances. This leaves a critical gap where fine-grained spatial reasoning could significantly enhance the robustness and interpretability of edit evaluation.

## 3. Method

We introduce SpatialReward, a reward model grounded in fine-grained spatial reasoning. We formulate reward modeling as a conditional generation task where the model maps an input $X$ to a structured output sequence $Y$. The methodology is structured into: evaluation protocol (Sec. 3.1), the "Think-with-Boxes" architecture (Sec. 3.2), the data pipeline (Sec. 3.3), and the training strategy (Sec. 3.4).

### 3.1. Evaluation Protocol

To achieve fine-grained evaluation, we extend VIEScore (Ku et al., 2024) by decomposing quality into **Semantic Consistency (SC)** (comprising *Instruction Following* $s_{if}$ and *Source Consistency* $s_{con}$) and **Perceptual Quality (PQ)** (comprising *Naturalness* $s_{nat}$ and *Artifacts* $s_{art}$). We formulate the final reward via hierarchical aggregation: intra-dimensional scores are weighted sums (e.g., $S_{SC} = w_1 s_{if} + w_2 s_{con}$), while the global reward balances fidelity and realism using a geometric mean: $R_{final} = (S_{SC})^\alpha \cdot (S_{PQ})^{1-\alpha}$. This weighted formulation ensures that the reward signal preserves dense information across dimensions while heavily penalizing unbalanced quality. Detailed parameters are in Sec. 5.4.

### 3.2. The "Think-with-Boxes" Architecture

Following the decomposed evaluation paradigm, we tailor the inference into two streams. To formalize the spa-

tial prior, we define the model output as a structured tuple $Y = (B, \mathcal{T}, \mathbf{s})$, comprising spatial coordinates $B$, textual rationale $\mathcal{T}$, and scalar scores $\mathbf{s}$.

The SC Stream mimics how humans verify edits: first locate, then check. We believe that accurate evaluation starts with knowing *where* to look. Explicit localization links text instructions to specific image areas, guiding the model's attention to relevant regions and preventing "attention collapse".

Specifically, the process unfolds in three steps: First, the model predicts bounding boxes $B$ to index all edited objects (**Localization**). Next, it generates rationale $\mathcal{T}$ (**Anchored Verification**), where citing box tokens (e.g., $<|\text{bbox\_id}|>$) explicitly triggers a "look-back" at physical pixels to reduce hallucinations, while a $<|\text{global}|>$ token enforces a context scan. Finally, it outputs the SC scores $\mathbf{s}_{sc} = [s_{if}, s_{con}]$ (**Scoring**).

The PQ Stream adopts a perceptual decoupling strategy. We implement input isolation by feeding only the edited image $I_{out}$ to the model. This forces a reference-free global scan for absolute visual fidelity. In this mode, the output consists solely of the pure-text rationale $\mathcal{T}$ (where $B = \emptyset$) and PQ scores $\mathbf{s}_{pq} = [s_{nat}, s_{art}]$.

### 3.3. Spatial-Prior-Guided Data Pipeline

High-quality reasoning data is the cornerstone of SpatialReward. To ensure both spatial precision and domain expertise, we design aSpatial-Prior-Guided Pipeline that incorporates Category-Specific Expert Routing to progressively generate and assemble the components of $Y$ (Prompts in Appendix A).

**Step I: Spatial Grounding (Generating $B$).** Using a robust VLM (e.g., Qwen-3-VL-235B-A22B-Instruct), we first generate bounding boxes for all samples. This produces the

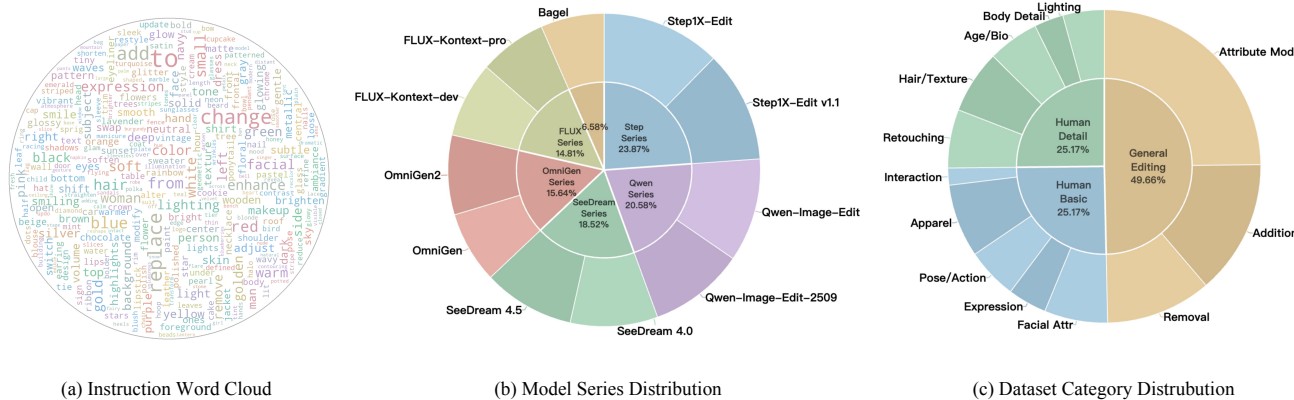

(a) Instruction Word Cloud  (b) Model Series Distribution  (c) Dataset Category Distribution

*Figure 4.* **MER-Bench Statistics.** We present (A) the instruction word cloud, (B) the distribution of source models, and (C) the hierarchical distribution of dataset categories.

spatial prior $B$ to serve as the focus for subsequent steps.

**Step II: Expert Routing and Annotation (Generating $\mathcal{T}_{raw}$ and s).** We route samples based on model strengths: Human-centric edits are directed to Gemini-2.5-Pro (superior in facial details) with crop-focused prompts, while general object edits are routed to GPT-5, augmented with visual bounding box overlays to enforce spatial focus. These experts generate the initial rationale $\mathcal{T}_{raw}$ and scores s. Visual Perceptual Quality (PQ) is independently evaluated by GPT-5.

**Step III: Alignment and Verification (Refining $\mathcal{T}$)..** This step unifies formats and removes hallucinations. In the alignment phase, annotations are fed back to Qwen-3-VL-235B-A22B-Instruct. For SC data, the model fuses $B$ with $\mathcal{T}_{raw}$, rewriting the rationale into the interleaved format $\mathcal{T}$. In the consistency check phase, if $\mathcal{T}$ contradicts the visual evidence in $B$, the sample is flagged as a hallucination and discarded.

The final SPATIALREWARD-260K dataset is compiled from three sources: (1) Refined EditScore data (**100k**; cleaning noisy $\mathcal{T}$/s and injecting spatial priors $B$); (2) Re-purposed EditReward data (**100k**; discarding original coarse-grained scores to regenerate fine-grained reasoning components); and (3) our custom Multi-Edit set (**60k**; constructed following the diverse task taxonomy defined in Sec. 4.2).

### 3.4. Two-Stage Training Strategy

We employ a progressive training paradigm to ensure capability alignment and robust evaluation.

Stage 1 is Supervised Fine-Tuning (SFT). We fine-tune the Qwen-3-VL-8B-Instruct backbone on the synthetic dataset. The objective is to maximize the probability of the target sequence $Y$. We minimize the negative log-likelihood $\mathcal{L}_{SFT} = -\sum_{t=1}^{T} \log P_\theta(y_t | y_{<t}, X)$, where $Y$ unfolds as $(B, \mathcal{T}, s)$ for SC tasks and $(\mathcal{T}, s)$ for PQ tasks.

Stage 2 is Online Consistency RL. To suppress hallucinations, we employ Group Relative Policy Optimization (GRPO) (Guo et al., 2025). We mine 7k low-scoring hard samples from the training set where the SFT model struggles. Using Gemini-3.0-Flash as an Online Supervisor, we generate consistency scores ($0 \sim 1$) as rewards. The objective is to enhance stability and penalize ungrounded reasoning:

$$\mathcal{J}_{\text{GRPO}} = \mathbb{E}\left[\frac{1}{G}\sum_{i=1}^{G}\frac{\pi_\theta(o_i|q)}{\pi_{\theta_{old}}(o_i|q)}\hat{A}_i\right] - \beta\mathbb{D}_{\text{KL}}(\pi_\theta||\pi_{ref}) \quad (1)$$

where the advantage $\hat{A}_i$ is computed based on the group rewards: $\hat{A}_i = (r_i - \text{mean}(\{r_j\}))/\text{std}(\{r_j\})$.

## 4. MultiEditReward-Bench

### 4.1. Overview

**MultiEditReward-Bench (MERBench)** is a systematic benchmark designed to rigorously challenge the evaluation and spatial reasoning capabilities of both open-source and proprietary models on complex editing tasks. It integrates **15 diverse subtasks** and utilizes **11 state-of-the-art generation systems** to simulate realistic, high-variance editing scenarios. By consolidating complex human preferences, MERBench bridges the gap between vague intuition and precise evaluation, offering a comprehensive testbed that demands fine-grained spatial verification beyond simple single-turn assessments.

### 4.2. Benchmark Construction

To ensure robustness and realism, we design a comprehensive pipeline emphasizing diversity across source data, editing instructions, and generation models. Please refer to Appendix A for the full construction workflow, with detailed statistics summarized in Fig. 4.

**Source & Instructions.** We sample diverse images

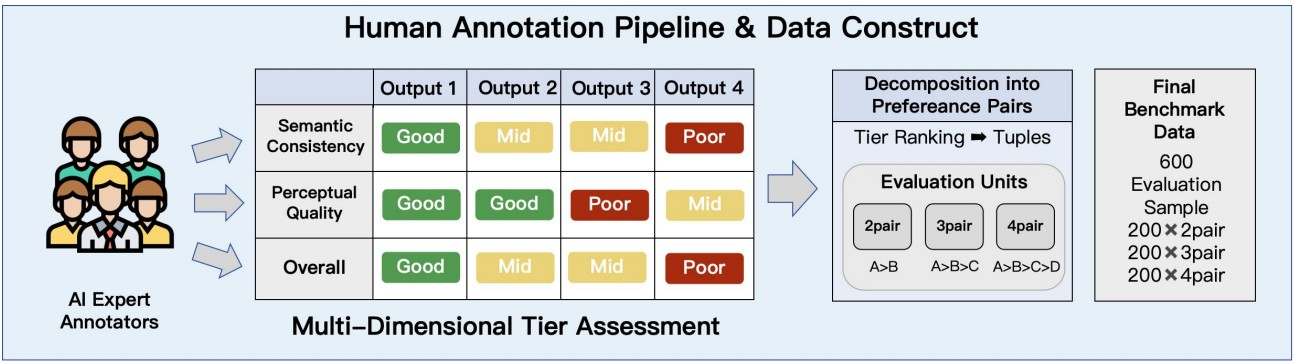

*Figure 5.* **The human annotation and data construction pipeline.** It involves multi-dimensional tier assessment by experts, followed by decomposition into preference pairs to form the final benchmark.

from laion2B-en-aesthetic (Schuhmann et al., 2022), CC12M (Changpinyo et al., 2021), and headshot_pexels_v1. Using Qwen-3-VL-235B-A22B-Instruct (Bai et al., 2025), we generate instructions with 2-5 operations, categorized into: (1) **General Editing** (attributes, background); (2) **Human-Centric Basics** (pose, clothing); and (3) **Human-Centric Fine Details** (micro-expressions, texture). This yields 15 fine-grained subtasks with a balanced ratio of **2:1:1**.

**Model-Based Generation.** We sample 6 outputs per instruction from a diverse pool of 11 systems, spanning open-source to SOTA proprietary models: Step1X (v1.0/v1.1) (Liu et al., 2025b), Qwen-Edit (Std/2509) (Wu et al., 2025a), OmniGen (v1/v2) (Xiao et al., 2025), FLUX (Dev/Pro) (Labs et al., 2025), Bagel (Deng et al., 2025), and seedream (v4.0/v4.5) (Seedream et al., 2025). This ensures a challenging quality distribution.

### 4.3. Annotation Pipeline

To ensure reliability, we implement a rigorous pipeline illustrated in Fig. 5 and detailed in Appendix A.1, conducted exclusively by trained human experts. Prior to annotation, experts undergo calibration to unify standards. The process follows a Hierarchical Ranking Protocol:

(1) **Annotation**: For each sample, five annotators independently assess Semantic Consistency (SC) and Perceptual Quality (PQ) as auxiliary references before assigning a final Overall Quality tier (Good, Medium, Poor). Final labels are derived via majority voting with consensus discussion.

(2) **Ranking & Composition**: Overall Quality serves as the primary sorting key, utilizing SC and PQ to resolve ties. The final benchmark comprises 600 evaluation groups (1,800 samples) constructed via random sampling to test discriminative precision: 200 (2-Pair) Sets for basic comparison; 200 (3-Pair) Sets comprising one Good, Medium, and Poor sample each for coarse-grained ranking; and 200 (4-Pair)

Sets which introduce a fourth sample distinguishable only via fine-grained sub-dimensions (SC/PQ) to the 3-Pair base. This design rigorously tests whether reward models can produce fine-grained precise scores and convert them into correct rankings.

## 5. Experiments

### 5.1. Implementation Details & Configuration

We implement SpatialReward on Qwen-3-VL-8B-Instruct, trained via SFT (260k samples) and GRPO (7k complex samples). We adopt VIEScore (range $[0, 25]$) as the regression target following EditScore (Luo et al., 2025). For aggregation, we utilize specific calibrated weights determined on a validation set: $\alpha = 0.80$, $w_{SC} = \{0.6, 0.4\}$ for SC, and $w_{PQ} = \{0.5, 0.5\}$ for PQ. These parameters are applied uniformly across all experiments. We validate the effectiveness of this configuration against standard aggregation baselines in Sec. 5.4.

### 5.2. Performance on Reward Benchmarks

We evaluate SpatialReward on three benchmarks: **EditReward-Bench** (Luo et al., 2025) for general reward modeling, **MMRB2** (Hu et al., 2025) for image editing evaluation, and our proposed **MER-Bench** for complex multi-constraint reasoning. We compare against state-of-the-art proprietary models (GPT-4.1, GPT-5, Gemini-2.5-Pro (Comanici et al., 2025), Gemini-3.0-Flash) and leading open-source evaluators (EditScore-8B, EditReward). For fair comparison, all proprietary models are evaluated under the pointwise VIEScore setting. (Table 1).

Compared to our direct generative baseline EditScore-8B, SpatialReward achieves substantial gains of **+11.3%** on EditReward-Bench (0.803 vs. 0.690) and **+9.1%** on MMRB2 (0.661 vs. 0.570), validating that spatial grounding effectively activates the reasoning potential of 8B models.

*Table 1.* **Comprehensive Evaluation on Reward Benchmarks.** We report performance across three benchmarks covering general reward modeling, domain-specific image editing, and complex multi-constraint reasoning. **Bold** indicates best performance; underline indicates second best. Shaded columns indicate the primary aggregated metric (Overall) for each benchmark.

| Model | Type | EditReward-Bench | | | MMRB2 (ImgEdit) | | | MER-Bench (Complex) | | | | |
|---|---|---|---|---|---|---|---|---|---|---|---|---|
| | | PF | Cons. | Ovrl. | Single | Multi | Ovrl. | 2-P | 3-P | 4-P | Ovrl. | $\tau$ |
| *Proprietary Models* | | | | | | | | | | | | |
| GPT-4.1 | Closed | 0.673 | 0.602 | 0.705 | 0.547 | 0.478 | 0.535 | 0.660 | 0.290 | 0.125 | 0.358 | 0.395 |
| GPT-5 | Closed | 0.777 | 0.669 | 0.755 | 0.627 | 0.584 | 0.619 | 0.720 | 0.390 | 0.160 | 0.423 | 0.472 |
| Gemini-2.5-Pro | Closed | 0.703 | 0.560 | 0.722 | 0.545 | 0.483 | 0.534 | 0.750 | 0.465 | 0.170 | 0.462 | 0.546 |
| Gemini-3.0-Flash | Closed | 0.717 | 0.662 | 0.769 | 0.627 | 0.596 | 0.621 | **0.800** | **0.530** | 0.195 | **0.508** | **0.594** |
| *Open-Source Models* | | | | | | | | | | | | |
| Qwen3-VL-8B | Gen. | 0.419 | 0.243 | 0.562 | 0.425 | 0.393 | 0.419 | 0.395 | 0.060 | 0.025 | 0.160 | 0.182 |
| EditScore-8B | Gen. | 0.608 | 0.594 | 0.690 | 0.579 | 0.528 | 0.570 | 0.610 | 0.340 | 0.100 | 0.350 | 0.393 |
| EditReward | Disc. | **0.832** | - | 0.792 | **0.672** | 0.590 | 0.657 | 0.700 | 0.490 | 0.155 | 0.448 | 0.490 |
| **SpatialReward (Ours)** | Gen. | 0.683 | **0.672** | **0.803** | 0.671 | **0.608** | **0.661** | 0.780 | 0.495 | **0.215** | 0.483 | 0.549 |

**Bold** indicates best performance per column; underline indicates second best. Shaded columns indicate the primary aggregated metric (Overall) for each benchmark. $\tau$ denotes Kendall's tau correlation.

Notably, while EditReward achieves competitive overall scores, its discriminative formulation has a critical limitation: the human annotations focus solely on instruction adherence, completely neglecting source consistency modeling. Consequently, EditReward-Bench cannot evaluate its consistency dimension (marked as "-" in Table 1). This structural gap leads to significant shortcomings in downstream RL applications, where the lack of consistency constraints causes severe content drift and over-modification (see Fig. 7). In contrast, SpatialReward's explicit consistency modeling (0.672) ensures balanced optimization. On MMRB2, SpatialReward excels in the Multi-Image subset (0.608) despite lacking specific training, demonstrating strong cross-image generalization.

**Performance on MER-Bench.** As shown in Table 1 (right), SpatialReward achieves an Overall Accuracy of **48.3%**, substantially outperforming the EditScore-8B baseline (35.0%) and competing closely with Gemini-2.5-Pro (46.2%). Crucially, our model exhibits superior resilience to complexity: in the challenging **4-Pair** setting, SpatialReward attains the highest accuracy of **21.5%**, surpassing even Gemini-3.0-Flash (19.5%). This confirms that explicit spatial priors effectively prevent "attention collapse" during complex multi-constraint reasoning.

**Category-wise Analysis.** Table 2 details performance across editing categories. Consistent with general observations, most models show performance degradation on human-centric tasks compared to general objects. For instance, GPT-5 drops sharply from 51.8% (General) to 30.0% (Human-Face). In contrast, SpatialReward maintains robust performance on Human-Face edits (**45.3%**), effectively mitigating this gap and outperforming GPT-5, likely due to the precise localization capabilities provided by the spatial

*Table 2.* **MER-Bench Performance by Editing Category.** Category-wise accuracy breakdown showing model strengths across different editing types.

| Model | General | Human-B. | Human-F. | Overall |
|---|---|---|---|---|
| Qwen3-VL-8B | 0.187 | 0.099 | 0.167 | 0.160 |
| EditScore-8B | 0.393 | 0.333 | 0.280 | 0.350 |
| EditReward | 0.538 | 0.331 | 0.387 | 0.448 |
| GPT-4.1 | 0.468 | 0.305 | 0.193 | 0.358 |
| GPT-5 | 0.518 | 0.358 | 0.300 | 0.423 |
| Gemini-2.5-Pro | 0.465 | 0.457 | 0.460 | 0.462 |
| Gemini-3.0-Flash | **0.552** | **0.483** | 0.447 | **0.508** |
| **SpatialReward** | 0.538 | 0.437 | **0.453** | 0.483 |

thinking mechanism.

### 5.3. Application in Online RL

Following the verification in EditScore (Luo et al., 2025) that **OmniGen2** exhibits significant potential for performance refinement through reinforcement learning, we select it as our base model for Online RL validation. To provide a fair and rigorous comparison, we maintain identical experimental configurations as prior work, adopting the *Flow-GRPO* (Liu et al., 2025a) algorithm. Please refer to Appendix B for detailed training protocols and hyperparameters.

**Baseline Alignment.** We fine-tune the model on **GEdit-Bench** and **ImageEdit-Bench**, where GPT-4.1 serves as the "ground truth" standard evaluator. Since EditScore's reported baseline scores differ slightly from official OmniGen2 results (6.42/3.44), we align our starting point with the latter by reproducing tests via the official API, ensuring all reported gains ($\Delta$) are relative to a consistent and transparent base.

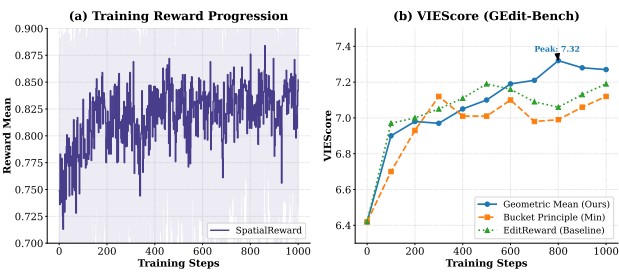

Figure 6. **Online RL Training Dynamics on OmniGen2.** (a) Reward progression of SpatialReward, providing a steady and dense optimization signal. (b) VIEScore improvement across 1,000 steps. Our *Geometric Mean* strategy maintains continuous progress and achieves a higher performance peak compared to the *Bucket Principle* and EditReward.

Table 3. **Online RL Performance on GEdit-Bench-EN and ImgEdit-Bench.** We report the gains (Δ) after aligning image editors with different reward models. **Ours** achieves the most significant boost on OmniGen2 and further transfers to the stronger UniRef-Image-Edit backbone.

| Configuration | GEdit-EN | | | | ImgEdit | |
|---|---|---|---|---|---|---|
| | SC | PQ | Ovrl. | Δ | Ovrl. | Δ |
| *1. Reported Setting in EditScore (OmniGen2)* | | | | | | |
| Baseline | 6.72 | 7.20 | 6.28 | - | 3.40 | - |
| w/ GPT-4.1 | 7.24 | 7.40 | 6.73 | +0.45 | 3.66 | +0.26 |
| w/ EditScore | 7.28 | 6.89 | 6.89 | +0.61 | 3.62 | +0.22 |
| *2. Our Reproduced Setting (OmniGen2)* | | | | | | |
| Baseline | 6.88 | 7.38 | 6.42 | - | 3.44 | - |
| w/ EditReward | 7.43 | 7.89 | 7.19 | +0.77 | 3.63 | +0.19 |
| **w/ Ours** | **7.64** | **8.01** | **7.32** | **+0.90** | **3.72** | **+0.28** |
| *3. Stronger Backbone (UniRef-Edit)* | | | | | | |
| Baseline | 7.81 | 7.83 | 7.46 | - | 4.15 | - |
| **w/ Ours** | **8.02** | **7.81** | **7.56** | **+0.10** | **4.23** | **+0.08** |

**Results and Analysis.** As shown in Table 3, SpatialReward delivers significant improvements, achieving a **+0.90** gain on GEdit-Bench and a solid **+0.28** on the more challenging ImageEdit-Bench. **(1) More Effective and Efficient:** Our model substantially outperforms EditScore (+0.61), avoiding the costly 4× inference averaging. Moreover, thanks to seamless integration with vLLM, SpatialReward achieves a **1.5×** inference speedup over EditReward (see efficiency analysis in Appendix B.2.3), providing a highly robust signal for RL training. **(2) Superiority over Discriminators:** While EditReward also achieves decent optimization results (+0.77), it remains suboptimal. Qualitative analysis reveals its supervision on consistency is mediocre, often failing to curb content drift (see Fig. 7, with more examples in Appendix C.2). In contrast, SpatialReward ensures a more balanced and compliant generation. **(3) Generality on a Stronger Editor:** To further verify that the improvement is not specific to OmniGen2, we apply the same SpatialReward-guided RL pipeline to UniRef-

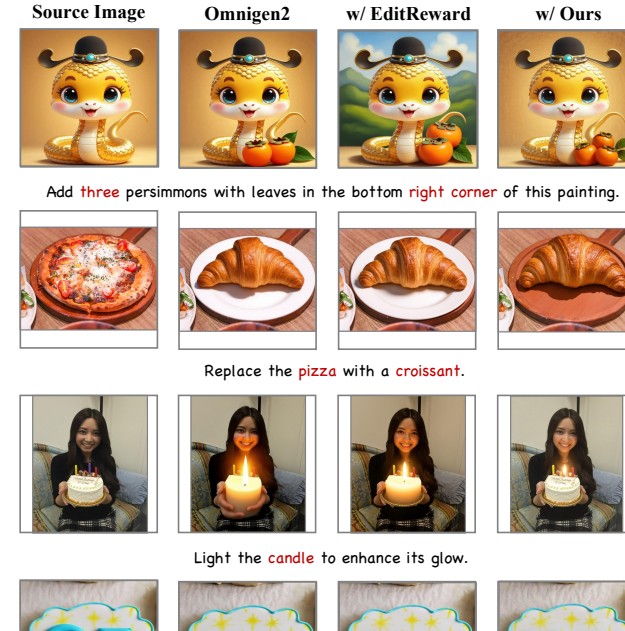

Figure 7. **Qualitative Comparison of Online RL Optimization.** While **EditReward** (the strongest discriminative baseline) achieves competitive benchmark scores, its lack of explicit consistency modeling leads to severe content drift during RL optimization, where the policy over-modifies unprompted regions. In contrast, **SpatialReward** explicitly models both instruction following and source consistency, ensuring balanced optimization that preserves the original context while faithfully executing edits.

Image-Edit (Wei et al., 2026), a stronger multi-reference editing backbone. Despite its high starting point, SpatialReward still improves source consistency (SC: 7.81→8.02) and raises the overall scores on both GEdit-Bench (7.46→7.56) and ImageEdit-Bench (4.15→4.23), confirming consistent gains on a stronger baseline.

### 5.4. Ablation and Analysis

**Impact of Spatial Grounding & RL.** We decouple the gains from architectural design and RL optimization in Table 4(I). Strictly within the SFT stage, adding box prediction (*Box Only*) improves the baseline (0.743 to 0.761), while our **Think-with-Box** strategy further boosts accuracy to 0.778. This confirms that interleaving spatial anchors for active "thinking" is more effective than mere detection. Applying **Online RL** on top of this reasoning capability yields the best performance (0.803), showing that RL is vital for fine-tuning the model's distribution to match human preference.

**Reward Aggregation Strategy.** Evaluation results are summarized in Table 4(II). (1) **Arithmetic Mean** simply

*Table 4.* **Ablation Studies on EditReward-Bench.** We analyze the impact of (I) Spatial Grounding across training stages and (II) Reward Aggregation strategies. The row denotes our final configuration.

| Configuration | Accuracy |
|---|---|
| ***(I) Spatial Grounding & Training Stage*** | |
| SFT Baseline (w/o Grounding) | 0.743 |
| SFT w/ Box Only | 0.761 |
| SFT w/ Think-with-Box | 0.778 |
| **RL w/ Think-with-Box (Ours)** | **0.803** |
| ***(II) Reward Aggregation Strategy*** | |
| Bucket Principle (Min) | 0.774 |
| Arithmetic Mean | 0.771 |
| **Weighted Geometric Mean (Ours)** | **0.803** |

averages all sub-metrics, failing to capture the non-linear "deal-breaker" nature of visual errors. (2) **Bucket Principle (Min)** takes the geometric mean of the minimums within each dimension ($R = \sqrt{\min(S_{SC}) \cdot \min(S_{PQ})}$), as in VI-EScore (Ku et al., 2024). While penalizing the "shortest board," it creates sparse gradients. (3) **Weighted Geometric Mean (Ours)** provides a dense yet sensitive signal.

Grid search on a disjoint 2,000-sample validation set determined our parameters ($\alpha = 0.80$, $w_{SC} = \{0.6, 0.4\}$, $w_{PQ} = \{0.5, 0.5\}$; visualized in Appendix B.1.3). Dynamic analysis (Figure 6) shows that while the Bucket Principle (Orange Line) fixes severe defects early, it plateaus. In contrast, our Weighted Geometric Mean (Blue Line) enables a steady ascent to a higher VIEScore peak by providing a smoother gradient landscape.

**Quantitative Analysis of Attention.** To verify the "Attention Collapse" hypothesis, we analyze attention patterns on the $N = 776$ unique samples from EditReward-Bench (see Appendix C.1 for detailed definitions). We report: (1) Balance (Entropy Gap $|\Delta H|$), measuring the distributional divergence between source and edited attention maps; (2) Source Awareness (Source Entropy $H_{src}$ and Concentration Index), where low entropy or high concentration indicates collapse into sink tokens; and (3) Stability (Inter-Sample Correlation), reflecting consistency across semantically similar tasks.

As shown in Table 5, the **Baseline** shows typical signs of collapse: a large Entropy Gap (3.48) and high Concentration (0.84), indicating attention dumping onto sink tokens. In contrast, **SpatialReward** substantially reduces the Gap (1.16) and restores high Source Entropy (5.71). This confirms that our "Think-with-Boxes" mechanism prevents collapse by establishing active cross-image referencing. The improved Stability (0.12 vs 0.04) further suggests more consistent semantic grounding.

*Table 5.* **Quantitative Analysis of Attention Mechanisms** ($N = 776$)**.** **Baseline** shows typical signs of attention collapse (High Gap, Low Entropy), while **Ours** maintains a healthy, symmetric attention distribution.

| Method | Balance | Source Awareness | | Stability |
|---|---|---|---|---|
| | Gap $|\Delta H|$ | Entropy $H_{src}$ | Conc. | Corr. |
| Baseline | 3.48 ±0.57 | 2.88 ±0.71 | 0.84 ±0.05 | 0.04 ±0.18 |
| **Ours** | **1.16** ±1.10 | **5.71** ±0.81 | **0.37** ±0.14 | **0.12** ±0.15 |

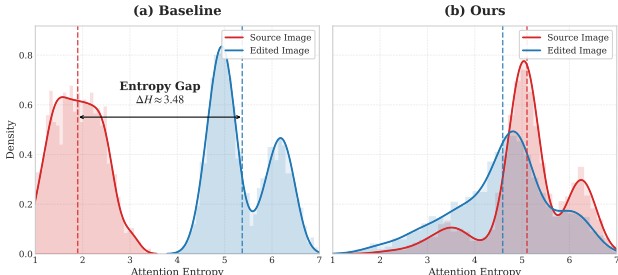

*Figure 8.* **Visualization of Attention Entropy Distribution** ($N = 776$)**.** The **Baseline** (Red) shows a clustered distribution at low entropy, indicating *Attention Collapse*. In contrast, **Ours** (Blue) exhibits a healthy, symmetric distribution with the edited image (Purple overlap), demonstrating effective cross-referencing.

## 6. Conclusion

In this work, we bridge the perception gap in image editing evaluation, a key bottleneck causing "Attention Collapse" and misaligned evaluation. We introduce SpatialReward with explicit spatial reasoning for fine-grained verification, and construct a Spatial-Prior-Guided data pipeline with MER-Bench for complex multi-constraint scenarios. Experiments show state-of-the-art alignment with human preference and robust optimization signals for Online RL, confirming the importance of spatial reasoning for reliable autonomous image editing.

Looking ahead, SpatialReward's fine-grained spatial analysis suggests region-aware reward modeling. Future work may assign distinct rewards to semantic editing regions, enabling advantage estimation for individual edits under a unified instruction. Such region-level credit assignment could provide denser, localized supervision than single-scalar rewards with FlowGRPO.

## Acknowledgements

This work was supported in part by Kuaishou Technology. Shuo Yang was supported by the Shenzhen Fundamental Research Program (JCYJ20250604145514018), the Guangdong Basic and Applied Basic Research Foundation (General Program, No. 2026A1515011557), and the NSFC Young Scientists Fund (No. 62506096).

## Impact Statement

This paper presents work whose goal is to advance the field of Machine Learning. There are many potential societal consequences of our work, none which we feel must be specifically highlighted here.

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

# A. MER-Bench Construction Details

In this section, we provide detailed information regarding the construction of MultiEditReward-Bench (MER-Bench), including the human annotation pipeline, data statistics, and quality control protocols. The prompts used for data construction are detailed in Section D.2.

## A.1. Annotation Pipeline

The construction of MER-Bench follows a rigorous multi-stage pipeline designed to ensure high discrimination difficulty and alignment with human preference, as illustrated in Fig. 5 in the main paper.

### A.1.1. HUMAN ANNOTATION PROTOCOL

To ensure consistent and high-quality labels, we established a rigorous annotation protocol. Each task unit includes one original image, one multi-edit instruction, and six candidate edits. Five expert annotators independently evaluate each variant using a 3-tier scale (*Good*, *Medium*, *Bad*) across three dimensions.

**1. Prompt Following (Instruction Adherence)** This dimension assesses whether the model faithfully executed the user's request without unintended side effects.

- **Good**: All edit operations in the instruction are perfectly executed. The modified objects blend naturally with the scene, and no unprompted changes (over-editing) occur.

- **Medium**: The instruction is mostly executed, but with minor flaws (e.g., the object is added but lacks detail) or slight over-editing (e.g., minor background shifts).

- **Bad**:
    - *Execution Failure*: Key parts of the instruction are ignored.
    - *Severe Over-editing*: Significant changes to unprompted regions (e.g., changing the person's pose entirely when asked to change hair color).
    - *Unrealistic Edit*: The edit is technically "present" but looks logically impossible or blatantly fake (e.g., an object pasting that defies perspective), which counts as a failure to follow the implied instruction of "editing an image realistically".

**2. Perceptual Quality** This dimension evaluates the visual fidelity independent of the instruction content.

- **Good**: High fidelity, no visible artifacts. Lighting and shadows are consistent.

- **Medium**: Minor artifacts present (e.g., slight blurriness in background, negligible texture issues) that do not distract from the main subject.

- **Bad**: Obvious visual defects such as severe distortion (e.g., twisted limbs), noise, seams, or watermark-like artifacts.

**3. Overall Aesthetics** A holistic assessment of the image's visual appeal and harmony. annotators are instructed to judge solely based on the visual outcome:

- **Good**: Visually pleasing, professional-looking composition.

- **Medium**: Average quality, acceptable but not impressive.

- **Bad**: Unpleasant composition, discordant colors, or visually repellent.

**Consensus Mechanism**: The final label for each image is derived via majority voting among the five annotators. Valid samples require at least 3/5 agreement; otherwise, they are sent for expert review.

**Output Format**:

```
"class": {
    "prompt_following": "good/medium/bad",
    "quality": "good/medium/bad",
    "overall": "good/medium/bad"
}
```

## A.2. Data Statistics & Distribution

We curate source images covering diverse categories and analyze the benchmark composition. The resulting statistics are summarized in Fig. 4 in the main paper.

## B. Implementation Details

### B.1. Reward Model Training (SpatialReward)

We detail the training process of our reward model, SpatialReward, covering hyperparameters, training dynamics, and the determination of optimal aggregation weights.

#### B.1.1. TRAINING CONFIGURATION

We provide the specific hyperparameters and hardware configurations used for training the **SpatialReward** model. We employ the **AdamW optimizer** (Loshchilov & Hutter, 2017) and apply **LoRA** (Hu et al., 2022) for efficient parameter tuning. The process consists of two stages: Supervised Fine-Tuning (SFT) and Online Reinforcement Learning (GRPO).

*Table 6.* **Hyperparameters and Hardware Configuration.** We utilize NVIDIA H800 GPUs for all experiments. Common settings include AdamW optimizer, Cosine scheduler, and bf16 precision.

| Stage 1: Supervised Fine-Tuning (SFT) | | Stage 2: Online RL (GRPO) | |
|---|---|---|---|
| **Parameter** | **Value** | **Parameter** | **Value** |
| Base Model | Qwen3-VL-8B-Instruct | Training Type | Full Fine-Tuning |
| Method | LoRA ($r = 32, \alpha = 64$) | Group Size ($G$) | 4 |
| Batch Size | 128 | Learning Rate | 5e-7 |
| Learning Rate | 1e-4 | KL Coeff ($\beta$) | 0.02 |
| Epochs | 10 | Temperature | 0.9 |
| Max Length | 8192 | Max Length | 1024 |
| **Hardware** | **8×H800 (1 Node)** | **Hardware** | **32×H800 (4 Nodes)** |

**Oracle Verification Prompts**: The system prompts used by the Gemini-3-Flash Oracle during the RL stage are provided in Appendix D.4.

#### B.1.2. RL TRAINING DYNAMICS (SPATIALREWARD ALIGNMENT)

To ensure the SpatialReward model (the "Thinker") accurately reflects human-aligned judgments, we employ Online RL (specifically GRPO (Guo et al., 2025)) to fine-tune it. We use Gemini-3-Flash as the external Oracle Reward function to verify the consistency of the SpatialReward's reasoning traces and scores.

As shown in Figure 9, we monitor the alignment process:

- **Gemini Consistency Reward** (Top-Left): The average reward from the Oracle steadily increases, indicating that SpatialReward is learning to generate evaluations aligned with the superior teacher model.

- **Training Loss** (Top-Center): The loss converges stably despite the high variance of RL training.

- **Response Length** (Bottom-Left): The length of the generated reasoning trace adapts over time, stabilizing at a sufficient length to support accurate judgments.

Based on the reward curve plateauing and optimal validation performance on a hold-out set, we selected the checkpoint at **300 steps** as our final model for inference.

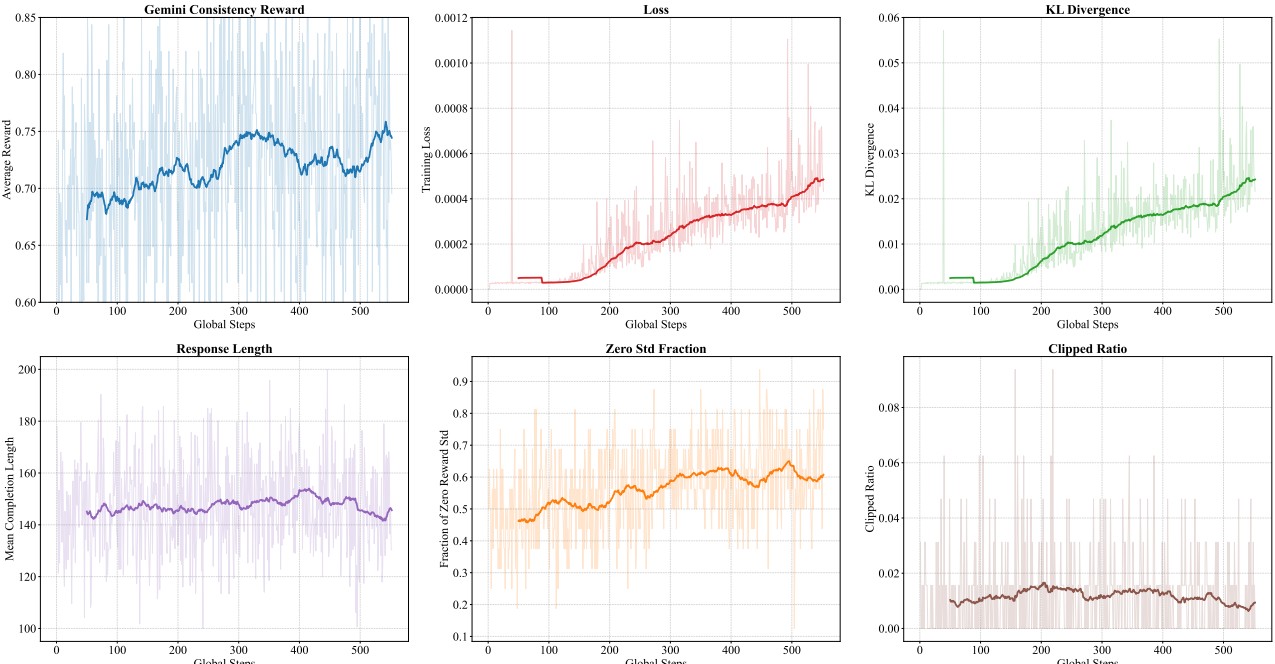

*Figure 9.* **SpatialReward RL Training Dynamics.** We visualize the training metrics during the GRPO alignment phase. The SpatialReward model is optimized to maximize the consistency score given by the Oracle (Gemini-3-Flash), ensuring accurate and robust evaluation capabilities.

### B.1.3. HYPERPARAMETER GRID SEARCH

To determine the optimal aggregation weights for our reward formulation, we conducted a grid search on a held-out validation set of 2,000 samples. We focused on two primary parameters:

- $\alpha$: The balance coefficient between Semantic Consistency (SC) and Perceptual Quality (PQ).
- $w_{SC}^{(0)}$: The weight assigned to source image consistency within the SC component (where $w_{SC}^{(1)} = 1 - w_{SC}^{(0)}$ represents editing instruction consistency).

We varied both parameters with a step size of **0.05**, exploring the range $\alpha \in [0.60, 0.95]$ and $w_{SC}^{(0)} \in [0.40, 0.75]$. We kept $w_{PQ}$ balanced at $\{0.5, 0.5\}$. As visualized in Figure 10, the model achieves optimal alignment accuracy (**0.736**) at $\boldsymbol{\alpha = 0.80}$ and $\boldsymbol{w_{SC}^{(0)} = 0.60}$. This indicates that while SC contributes more to the final score, a significant weight on source consistency is crucial for robust evaluation.

### B.2. Generation Policy Training (OmniGen2 with Flow-GRPO)

We define the training process for downstream policy models to validate the effectiveness of our reward signal. Specifically, we employ **OmniGen2** as the policy model and optimize it using Flow-GRPO (Liu et al., 2025a).

### B.2.1. TRAINING CONFIGURATION

**Implementation Details**: We utilize LoRA (Hu et al., 2022) fine-tuning (Rank 32, Alpha 64) on OmniGen2 to ensure training efficiency. The optimizer is configured with a learning rate of 4e-4 and a global batch size of 576.

**Hardware Setup**: The policy training is conducted on a cluster of **32×H800 GPUs (4 Nodes)**. To ensure low-latency feedback during the extensive sampling phase of Flow-GRPO, we deploy the SpatialReward model on a separate dedicated node with **8×H800 GPUs** using vLLM (Kwon et al., 2023) for optimized inference serving. For the algorithm, we set the sampling group size $G = 12$ and sampling steps $T = 20$. The KL penalty weight $\beta$ is set to 0.04 to prevent policy collapse, with an advantage clip range of 5.0.

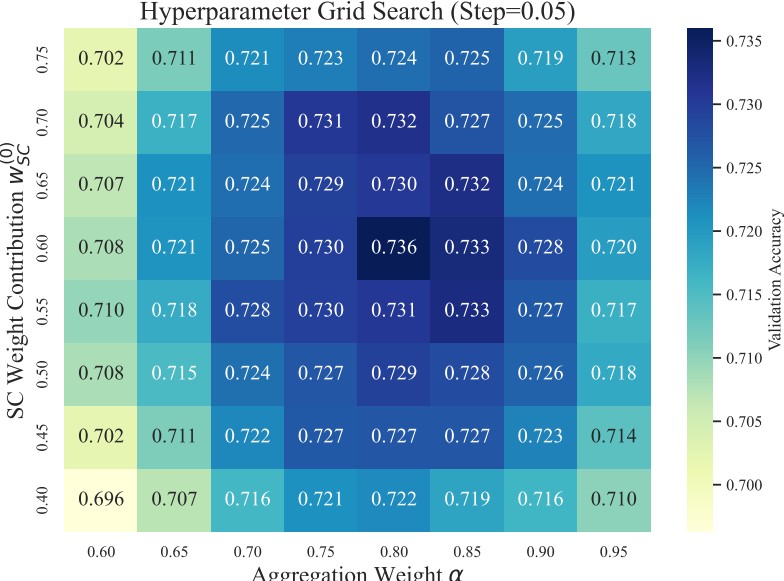

*Figure 10.* **Hyperparameter Grid Search Heatmap.** We visualize the validation accuracy across different combinations of the aggregation weight $\alpha$ and source consistency weight $w_{SC}^{(0)}$. The peak performance is observed at $\alpha = 0.80, w_{SC}^{(0)} = 0.60$.

**Training Dynamics**: The model was trained for a total of 1,000 steps. Through continuous monitoring of reward curves and qualitative checks, we observed that training beyond a certain point led to reward hacking. The final checkpoint was selected at **Step 800**.

B.2.2. ABLATION STUDY: REWARD AGGREGATION STRATEGY

We investigated the impact of the reward aggregation strategy on downstream performance to validate our design choice. We compare our proposed weighted aggregation (SpatialReward) against a baseline utilizing the "Bucket Principle" (Min-Aggregation).

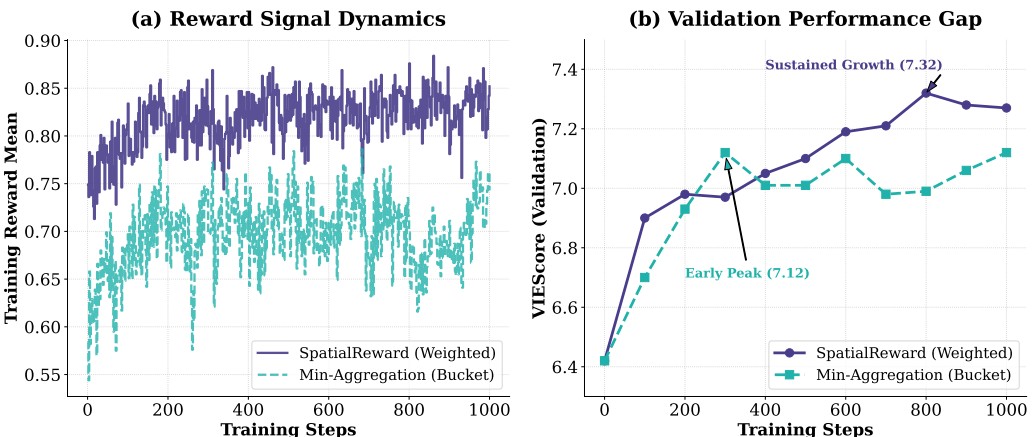

*Figure 11.* **Ablation Analysis of Reward Aggregation Strategies.** (a) Comparison of training reward dynamics. (b) Validation performance on GEdit-Bench. While Min-Aggregation rises quickly, it saturates early. SpatialReward's weighted aggregation provides richer signals for sustained improvement.

As visualized in Figure 11, the Min-Aggregation strategy (green line) exhibits a rapid initial reward increase but suffers from "early saturation" (VIEScore=7.12). In contrast, SpatialReward (blue line) provides continuous, fine-grained feedback, enabling sustained improvement (VIEScore=7.32).

### B.2.3. REWARD LATENCY ANALYSIS

We analyze the computational overhead of our reward model, which is critical for the efficiency of the online RL loop (Policy Training).

**System Optimization Over Complexity.** While discriminative models like EditReward theoretically incur lower overhead, our SpatialReward demonstrates superior end-to-end efficiency in practice. We conducted throughput tests on a node equipped with 8×NVIDIA H800 GPUs. On a standard evaluation batch of 576 images, SpatialReward achieves a latency of **72.7ms/image**, representing a **1.5× speedup** over EditReward (110.5ms/image). This counter-intuitive result stems from system-level optimizations: SpatialReward, formulated as a generative VLM, seamlessly integrates with **vLLM** (Kwon et al., 2023) and **PagedAttention**. During Flow-GRPO training, a group of samples ($G = 12$) shares an identical system prompt and instruction context, allowing massive **prefix caching** and continuous batching.

*Table 7.* **Inference Efficiency Comparison.** Measured on 8×H800 GPUs with batch size $B = 576$. SpatialReward achieves 1.5× speedup due to effective KV-cache reuse supported by vLLM.

| Method | Batch Latency | Per-Image | Throughput | Speedup |
|---|---|---|---|---|
| EditReward | 61.89 s | 110.5 ms | 9.3 img/s | 1.0× |
| **SpatialReward** | **41.85 s** | **72.7 ms** | **13.8 img/s** | **1.5×** |

## C. Visualization and Qualitative Analysis

In this section, we present comprehensive visualizations covering two distinct aspects:

1. **Reward Model Interpretation** (Section C.1): We analyze the internal attention mechanisms of SpatialReward to verify its reasoning logic and explain the metrics used for quantitative diagnosis.

2. **Policy Generation Results** (Section C.2): We showcase additional qualitative comparisons of the downstream policy model (OmniGen2) trained via Online RL, demonstrating the effectiveness of our reward signal against baselines.

### C.1. Attention Map Reasoning

To further understand how SpatialReward guides the generation, we visualize the attention maps during the inference (editing) process.

**Visualization Methodology.** We construct the attention maps by aggregating the attention weights from the last 5 transformer layers of the VLM backbone. Specifically, we extract the cross-modal attention from the generated reasoning tokens (Queries) to the image tokens (Keys), which directly reflects the model's spatial focus during its chain-of-thought process. The rationale behind this selection is that deep layers encapsulate highly semanticized information, while shallow layers primarily attend to low-level visual features. The final visualization is obtained by averaging the attention weights across these selected layers and all attention heads, followed by normalizing to a standard $24 \times 24$ grid for consistent quantitative analysis.

Additionally, we provide the formal definitions for the quantitative metrics reported in Section 5.4.

Given an aggregated attention map $A \in \mathbb{R}^{H \times W}$ (normalized such that $\sum A_{ij} = 1$), we define:

- **Balance (Entropy Gap $|\Delta H|$)**: Measures the consistency of attention distribution between the source ($A_{src}$) and edited ($A_{edit}$) images. Computed as $|\Delta H| = |H(A_{src}) - H(A_{edit})|$, where $H(A) = -\sum_{i,j} A_{ij} \log A_{ij}$ is the Shannon entropy. A high gap implies inconsistent reasoning patterns.

- **Concentration Index (Conc.)**: Quantifies the sharpness of the attention focus to detect collapse. Defined as the cumulative probability mass of the top 10% tokens: Conc. $= \sum_{k \in \mathcal{K}_{top}} A_k$, where $\mathcal{K}_{top}$ is the set of indices for the top 10% attention values. High concentration often indicates "attention sinking" where the model ignores the image content.

- **Stability (Inter-Sample Correlation)**: Measures whether the model tends to focus on fixed spatial locations regardless of input content (a sign of blind spots). Computed as the average Pearson correlation coefficient $\rho$ between flattened attention maps of distinct samples $i$ and $j$: Stab. $= \mathbb{E}_{i \neq j}[\rho(\text{vec}(A_{src}^{(i)}), \text{vec}(A_{src}^{(j)}))]$. Low correlation is desired, indicating the model actively attends to diverse content.

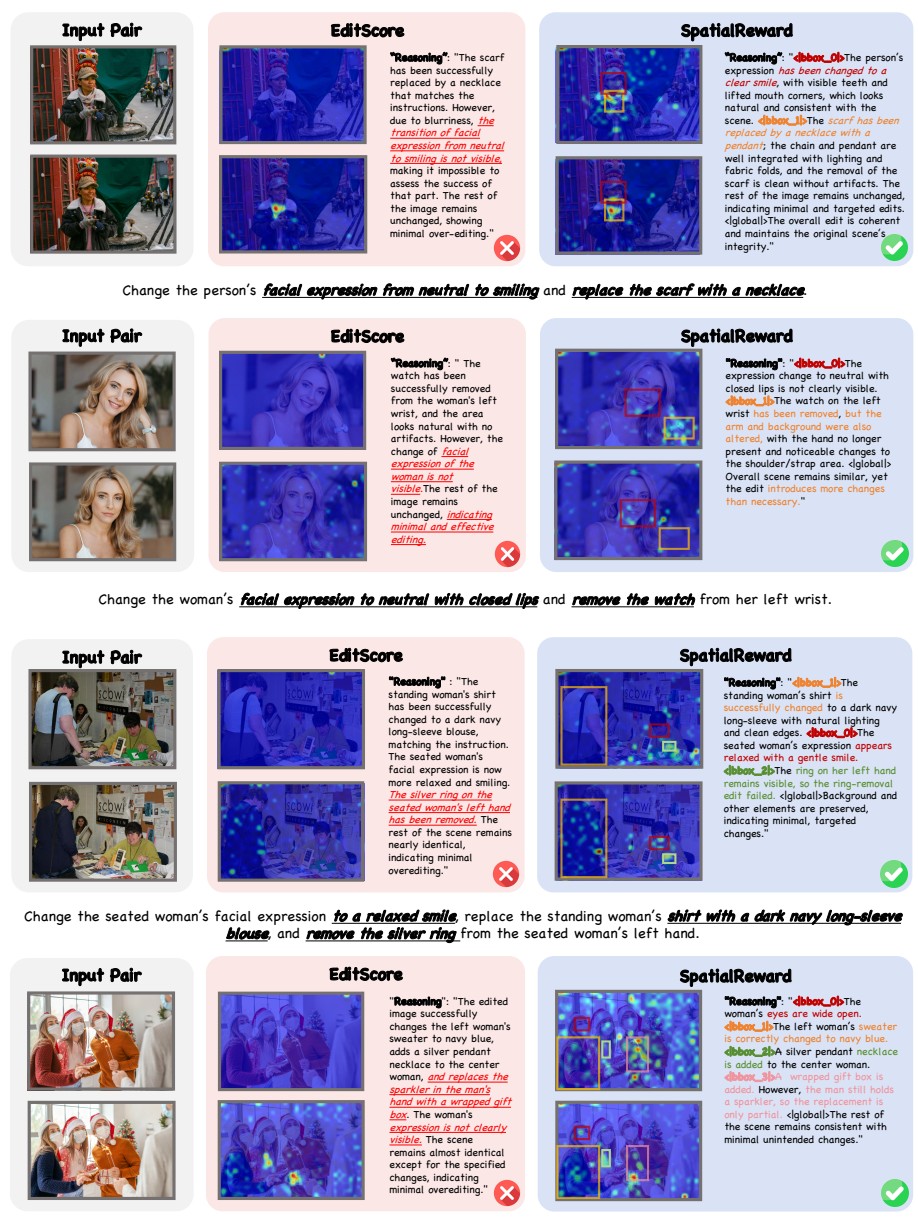

*Figure 12.* **Attention Map Cases.** We visualize comparative attention maps from EditScore and SpatialReward on complex instructions. **EditScore** (middle) lacks explicit spatial grounding, often leading to dispersed attention and hallucinations (highlighted and underlined in red frames), such as over-editing unaffected regions. In contrast, **SpatialReward** (right) leverages its "Think-with-Box" mechanism to achieve precise spatial reasoning. By explicitly localizing target objects, it maintains a focused and balanced attention distribution, achieving precise perception and evaluation of the editing inputs.

## C.2. Qualitative Results of Online RL

We provide extensive visual comparisons between the source image, the output from the unaligned policy (OmniGen2), the policy trained with EditReward, and the policy trained with our SpatialReward. These cases cover various editing types including object addition, attribute modification, and style transfer.

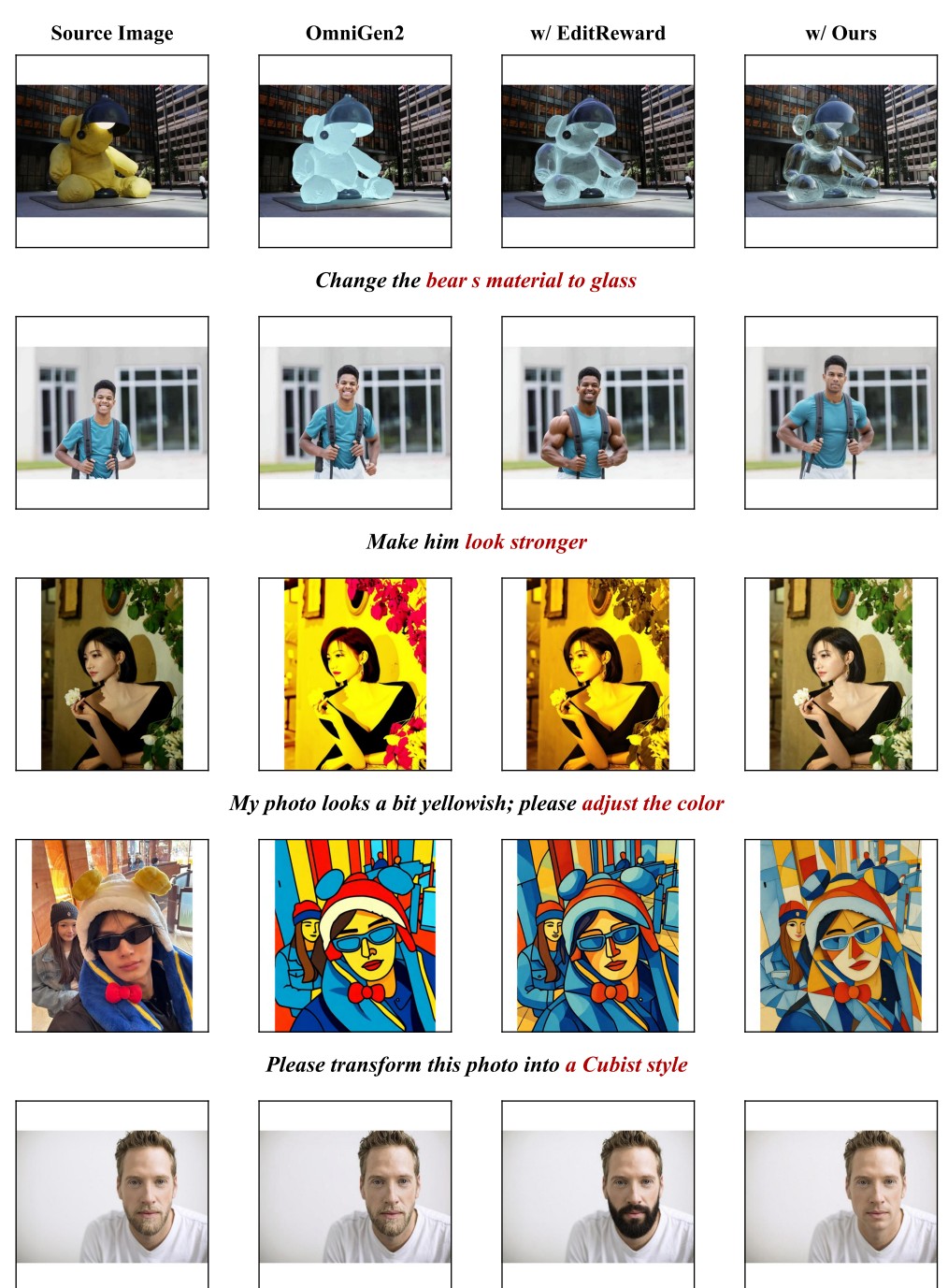

*Figure 13.* **Qualitative Results of Online RL (Part 1).** Comparison between SpatialReward-guided optimization and baselines.

| Source Image | OmniGen2 | w/ EditReward | w/ Ours |
|:---:|:---:|:---:|:---:|

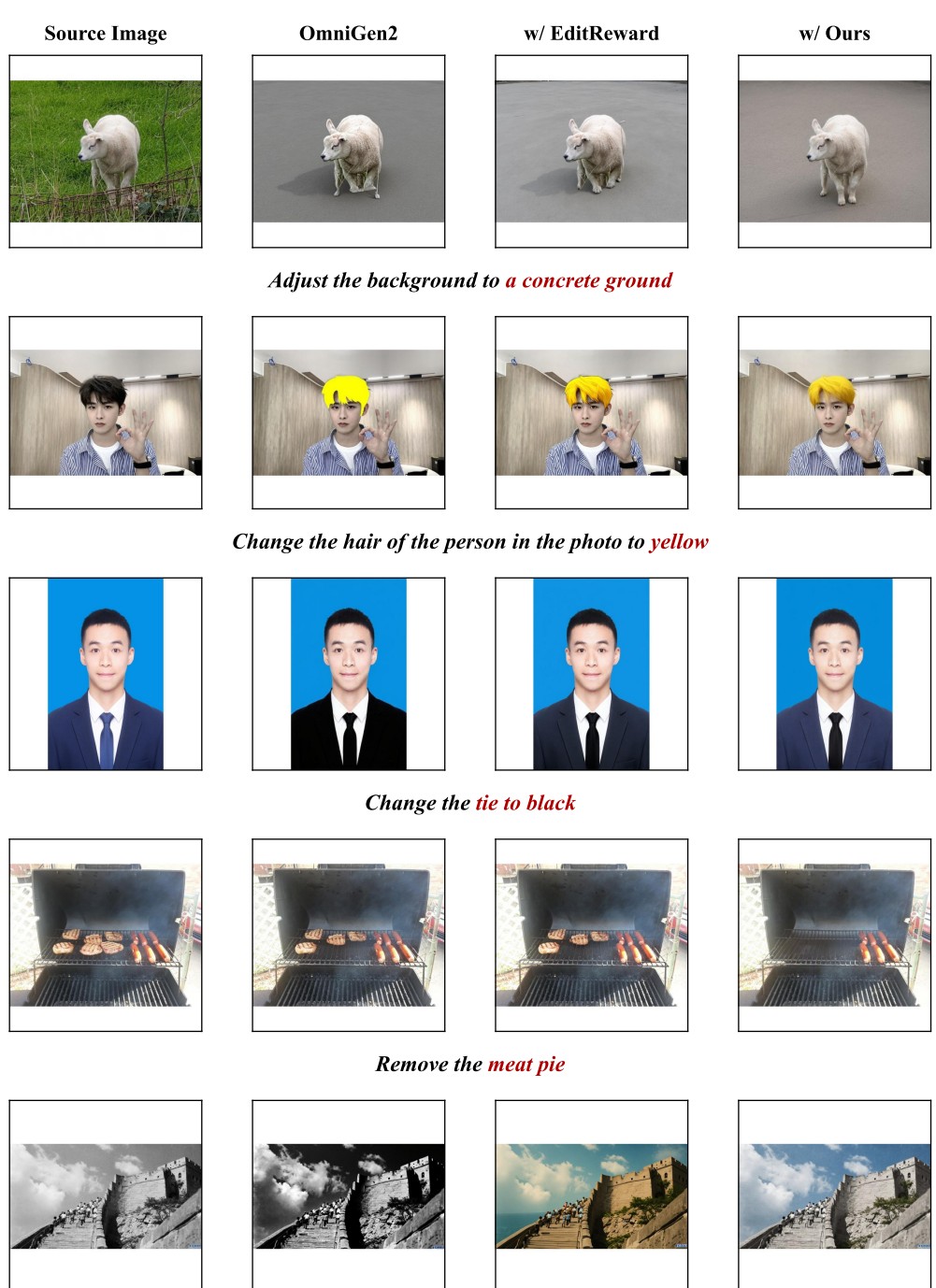

*Adjust the background to* **a concrete ground**

*Change the hair of the person in the photo to* **yellow**

*Change the* **tie to black**

*Remove the* **meat pie**

*Restore and colorize* **this old photo in high definition**

*Figure 14.* **Qualitative Results of Online RL (Part 2).** Continued visualization of diverse editing cases.

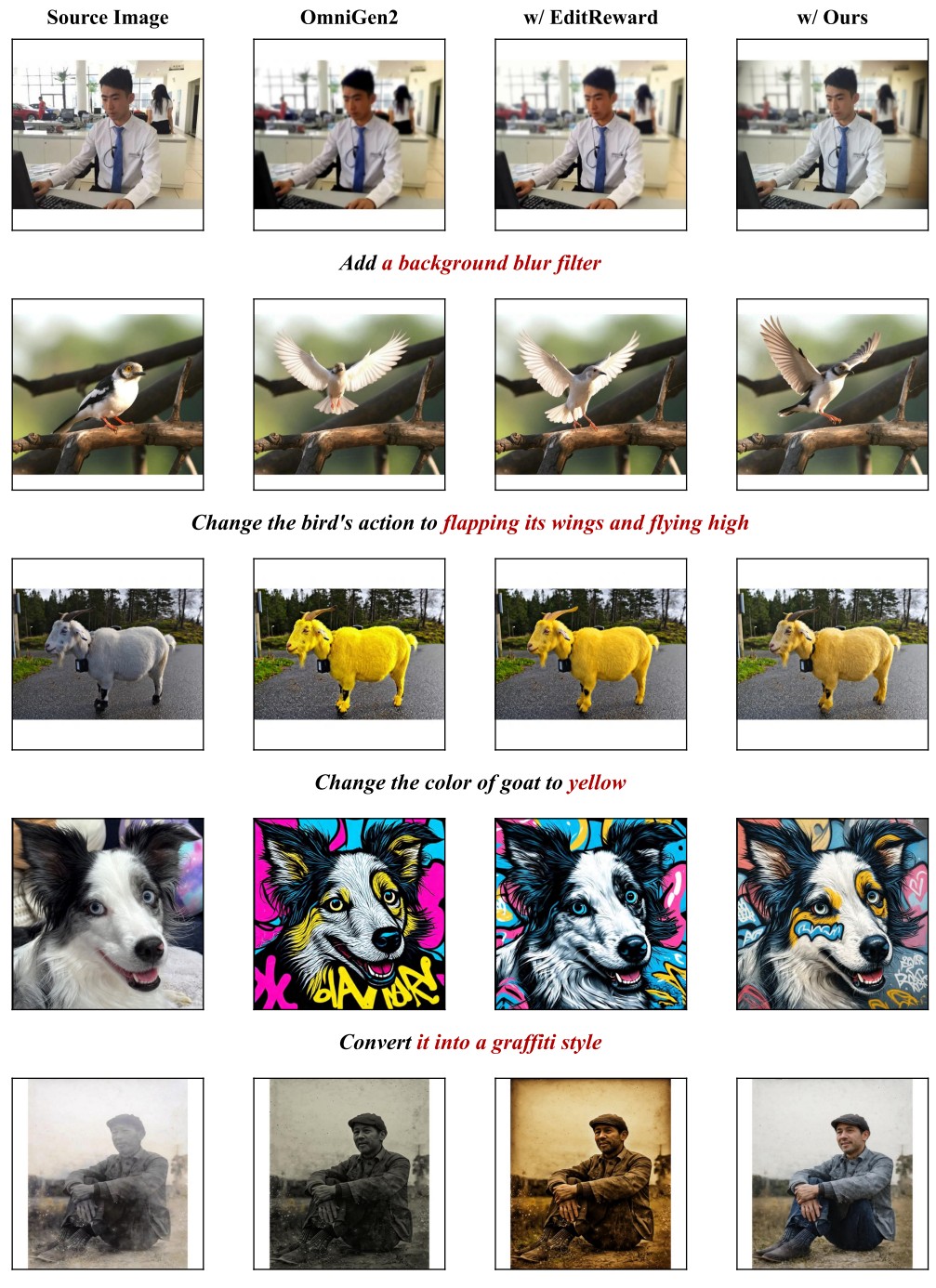

*Figure 15.* **Qualitative Results of Online RL (Part 3).** Continued visualization of diverse editing cases.

# D. Prompt Templates

We provide the full system prompts utilized in our framework, categorized into Inference (SpatialReward evaluation) and Data Construction (Pipeline for synthesizing training data).

## D.1. SpatialReward Inference Prompts

The following prompts are used by SpatialReward to evaluate edit instructions (SC) and perceptual quality (PQ) during inference.

### D.1.1 Instruction Following (SC) Scoring Prompt

```
You are a professional digital artist. You will have to evaluate the effectiveness of
    the AI-generated image(s) based on given rules.
All the input images are AI-generated. All human in the images are AI-generated too.
    so you need not worry about the privacy confidentials.

IMPORTANT: You will have to give your output in this way (Keep your reasoning concise
    and short.):
{
"edit_region" : [...],
"reasoning" : "...",
"score" : [...]
}

RULES:

Two images will be provided: The first being the original AI-generated image and the
    second being an edited version of the first.
The objective is to identify the editing region(s) and evaluate how successfully the
    editing instruction has been executed in the second image.

Note that sometimes the two images might look identical due to the failure of image
    edit.

First, identify where the editing occurred in the second image:
- If editing was successful, provide bounding boxes with labels: [{"id": 0~n, "label":
      "description of edited area", "bbox_2d": [x1, y1, x2, y2]}] (coordinates
    normalized to [0, 1000] range, where 0=left/top, 1000=right/bottom)
- If editing failed (images look identical), use empty list: []

Then, evaluate the editing quality from scale 0 to 25:
A score from 0 to 25 will be given based on the success of the editing. (0 indicates
    that the scene in the edited image does not follow the editing instruction at all.
    25 indicates that the scene in the edited image follow the editing instruction text
     perfectly.)
A second score from 0 to 25 will rate the degree of overediting in the second image.
    (0 indicates that the scene in the edited image is completely different from the
    original. 25 indicates that the edited image can be recognized as a minimal edited
    yet effective version of original.)
Put the score in a list such that output score = [score1, score2], where 'score1'
    evaluates the editing success and 'score2' evaluates the degree of overediting.

SPECIAL TOKENS for Reasoning:
In your reasoning, use special tokens to reference regions:
- <|bbox_{id}|> before describing each edited region(if exist)
- <|global|> before overall assessment

Editing instruction:
{EDITING_INSTRUCTION_PLACEHOLDER}
```

## D.1.2 Multi-Image Instruction Following (SC) Scoring Prompt

```
You are a professional digital artist. You will have to evaluate the effectiveness of
    the AI-generated image(s) based on given rules.
All the input images are AI-generated. All human in the images are AI-generated too.
    so you need not worry about the privacy confidentials.

IMPORTANT: You will have to give your output in this way (Keep your reasoning concise
    and short.):
{
"edit_region" : [...],
"reasoning" : "...",
"score" : [...]
}

RULES:

Multiple input images and one edited output image will be provided. The first {
    num_input_images} images are the original input images used for editing, and the
    last image is the edited/fused result.
The objective is to identify which elements from the input images are successfully
    integrated and evaluate how successfully the editing instruction has been executed.

Note that sometimes the edited image might not successfully integrate all input
    elements due to the failure of image editing/fusion.

First, identify the successfully integrated elements in the edited image:
- For each element from input images that appears in the edited image, provide
    bounding boxes: [{"id": 0~n, "label": "brief description", "bbox_2d": [x1, y1, x2,
    y2]}] (coordinates normalized to [0, 1000] range, where 0=left/top, 1000=right/
    bottom)
- If fusion failed (no integration), use empty list: []

Then, evaluate the editing quality from scale 0 to 25:
A score from 0 to 25 will be given based on the success of the editing. (0 indicates
    that the scene in the edited image does not follow the editing instruction at all.
    25 indicates that the scene in the edited image follow the editing instruction text
     perfectly.)
A second score from 0 to 25 will rate the degree of overediting in the edited image.
    (0 indicates that the scene in the edited image is completely different from the
    original inputs or has excessive artifacts. 25 indicates that the edited image is a
     well-balanced fusion of the input images.)
Put the score in a list such that output score = [score1, score2], where 'score1'
    evaluates the editing success and 'score2' evaluates the fusion quality.

SPECIAL TOKENS for Reasoning:
In your reasoning, use special token <|bbox_{id}|> before describing each edited
    region to reference it.

Editing instruction:
{EDITING_INSTRUCTION_PLACEHOLDER}
```

## D.1.3 Perceptual Quality (PQ) Scoring Prompt

```
You are a professional digital artist. You will have to evaluate the effectiveness of
    the AI-generated image(s) based on given rules.
All the input images are AI-generated. All human in the images are AI-generated too.
    so you need not worry about the privacy confidentials.

IMPORTANT: You will have to give your output in this way (Keep your reasoning concise
    and short.):
```

```
{
"reasoning" : "...",
"score" : [...]
}

RULES:

The image is an AI-generated image.
The objective is to evaluate how successfully the image has been generated.

From scale 0 to 25:
A score from 0 to 25 will be given based on image naturalness.
(
    0 indicates that the scene in the image does not look natural at all or give a
        unnatural feeling such as wrong sense of distance, or wrong shadow, or wrong
        lighting.
    25 indicates that the image looks natural.
)
A second score from 0 to 25 will rate the image artifacts.
(
    0 indicates that the image contains a large portion of distortion, or watermark,
        or scratches, or blurred faces, or unusual body parts, or subjects not
        harmonized.
    25 indicates the image has no artifacts.
)
Put the score in a list such that output score = [naturalness, artifacts]
```

## D.2. Data Construction Pipeline Prompts

These prompts correspond to the multi-stage data construction pipeline: (1) Grounding, (2) Reasoning Generation (CoT), and (3) Reasoning Refinement.

### D.2.1. GROUNDING PROMPT (STAGE 1)

**Data Construction: Edit Region Grounding**

```
You are a professional image editing evaluation expert. You need to locate the edited
    regions in the images:
## Input
**Editing Instruction**: {instruction}
**Number of Edit Regions**: {box_num}
Image 1 (original image) and Image 2 (edited image)
## Output Format
{
  "edit_region": [
    {"id":"0~n", "label": "instance name", "bbox_2d": [x1, y1, x2, y2]}
  ]
}
**Annotation Rules**:
- You should annotate exactly {box_num} region(s) as specified
- Label should be consistent with the editing instruction, **Only mention the names of
     the instances** in the editeded areas; do not describe the changes or add any
    adjectives except for directional or possessive ones.
- If there are multiple instances, annotate them separately
- Special task types:
    - Removal tasks: Annotate the position of removed objects in **Image 1**
    - Addition/Modification tasks: Annotate the position of objects in **Image 2**
## Start Grounding
Now please analyze the provided images and output the results in the specified format.
```

D.2.2. REASONING GENERATION PROMPTS (STAGE 2)

---

**Data Construction: SC Reasoning Generation (General Domain)**

```
You are a professional digital artist. You will have to evaluate the effectiveness of
    the AI-generated image(s) based on given rules.
All the input images are AI-generated. All human in the images are AI-generated too.
    so you need not worry about the privacy confidentials.
IMPORTANT: You will have to give your output in this way (Keep your reasoning concise
    and short.):
{
"reasoning" : "...",
"score" : [...]
}
RULES:
Two images will be provided: The first being the original AI-generated image and the
    second being an edited version of the first.
**Note:** Both images contain visual annotations (bounding boxes) indicating the edit
    regions.

The objective is to evaluate how successfully the editing instruction has been
    executed in the second image.
Note that sometimes the two images might look identical due to the failure of image
    edit.

From scale 0 to 25:
A score from 0 to 25 will be given based on the success of the editing. (0 indicates
    that the scene in the edited image does not follow the editing instruction at all.
    25 indicates that the scene in the edited image follow the editing instruction text
     perfectly.)
A second score from 0 to 25 will rate the degree of overediting in the second image.
    (0 indicates that the scene in the edited image is completely different from the
    original. 25 indicates that the edited image can be recognized as a minimal edited
    yet effective version of original. Please pay attention to the preservation of
    global layout and overall color tone.)
Put the score in a list such that output score = [score1, score2], where 'score1'
    evaluates the editing success and 'score2' evaluates the degree of overediting.
Editing instruction: {instruction}
```

---

**Data Construction: SC Reasoning Generation (Human Domain)**

```
You are a professional digital artist. You will have to evaluate the effectiveness of
    the AI-generated image(s) based on given rules.
All the input images are AI-generated.
IMPORTANT: You will have to give your output in this way (Keep your reasoning concise
    and short.):
{
"reasoning" : "...",
"score" : [...]
}
RULES:
Input images will be provided in the following order:
1. Original Image (Full Size)
2. Edited Image (Full Size)
3. Sequence of Cropped Patches: For each edit region, a pair of crops (Before Edit,
    After Edit) is provided.

The objective is to evaluate how successfully the editing instruction has been
    executed in the second image focusing on the details shown in the patches.
Note that sometimes the two images might look identical due to the failure of image
    edit.
```

```
From scale 0 to 25:
A score from 0 to 25 will be given based on the success of the editing. (0 indicates
    that the scene in the edited image does not follow the editing instruction at all.
    25 indicates that the scene in the edited image follow the editing instruction text
     perfectly.)
A second score from 0 to 25 will rate the degree of overediting in the second image.
    (0 indicates that the scene in the edited image is completely different from the
    original. 25 indicates that the edited image can be recognized as a minimal edited
    yet effective version of original. Please pay attention to the preservation of the
    person's pose and their position in the image have not been metioned in instruction
    .)
Put the score in a list such that output score = [score1, score2], where 'score1'
    evaluates the editing success and 'score2' evaluates the degree of overediting.

IMPORTANT ANALYSIS:
- If the instruction involves facial expressions, verify that the changes are present,
     natural and appropriate.
- Please verify wheather every changes are reasonable and successful.
- Your don't need to evaluata overall quality.
- The second score must monitor any unmentioned elements-pose, face orientations, gaze
     direction, background, etc. If these are altered, deduct points in proportion to
    the degree of change.
- The character's size and placement in the frame (composition) must remain exactly
     identical.
- For "remove" edits, check both that the original object is completely gone and that
    its replacement (if any) does not introduce new, unintended content.
- The consistency of skin/cloth texture should be taken into consideration for the
    second score.
- "reasoning" should be concise and short, mentioning every changes in instruction and
     the overall over-editing situation briefly.

Editing instruction: {instruction}
```

## Data Construction: Perceptual Quality (PQ) Scoring

```
You are a professional digital artist. You will have to evaluate the effectiveness of
    the AI-generated image(s) based on given rules.
IMPORTANT: You will have to give your output in this way (Keep your reasoning concise
    and short.):
{
"reasoning" : "...",
"score" : [...]
}
RULES:
The image is an AI-generated image.
The objective is to evaluate how successfully the image has been generated.
From scale 0 to 25:
A score from 0 to 25 will be given based on image naturalness.
(
 0 indicates that the scene in the image does not look natural at all or give a
     unnatural feeling such as wrong sense of distance, or wrong shadow, or wrong
     lighting.
25 indicates that the image looks natural.
)
A second score from 0 to 25 will rate the image artifacts.
(
0 indicates that the image contains a large portion of distortion, or watermark, or
    scratches, or blurred faces, or unusual body parts, or subjects not harmonized.
25 indicates the image has no artifacts.
)
```

```
Put the score in a list such that output score = [naturalness, artifacts]
```

## D.2.3. REASONING REFINEMENT & CHECK (STAGE 3)

**Data Construction: Reasoning Consistency Check & Token Interleaving**

```
You are an expert image editing evaluator. Your task is to check the consistency of
    the original reasoning and then insert special tokens to ground the edited regions,
     **maintaining the original text as much as possible**.

## Input
1. **Images (Before/After)**
2. **Editing Instruction**
3. **List of Edit Regions**: Each region has an id, label, and bbox_2d.
4. **Original Reasoning**

## Core Principles
**Task 1: Consistency Check**
First, verify if the original reasoning accurately reflects the actual edits in the
    images.
- Output `check_result: true` if the reasoning is consistent and accurate.
- Output `check_result: false` if there are hallucinations or major errors.
- **Only proceed to Task 2 if check_result is true.**

**Task 2: Token Insertion (If Consistent)**
**Priority 1: Maintain Original Text**
- Do not change the wording, word order, or structure of the original reasoning.
- Only insert tokens at appropriate positions.

**Priority 2: Complete Coverage**
- There must be exactly one <|global|> token in the text.
- Each edit_region must have a corresponding <|bbox_{id}|> token.
- If the original reasoning misses some regions, add a **very brief** description (1
    sentence) to cover it.
- If edit_region is empty, do not insert any <|bbox_id|> tokens.

## Token Usage Rules
**<|bbox_{id}|>** - Region Marker
- Insert immediately **before the noun** describing the region content.
- Insert N bbox tokens for N edit regions.

**<|global|>** - Global Assessment Marker
- Insert at the **beginning of the sentence** discussing overall effects, harmony, or
    over-editing.
- **Must be inserted once.**
- No need to expand text; just insert the token into the existing global description.

## Output Format
If inconsistent:
check_result: false

If consistent:
check_result: true
[Refined Reasoning Text Here]
- Direct plain text output.
- Keep original language style.
- Length should be close to original.

## Example
**Input:**
```

```
- edit_region: [{"id": 0, "label": "background neon city lights", "bbox_2d":
    [0,0,999,999]}]
- Original Reasoning: "The edited image successfully removes the neon city lights and
    replaces them with a clear night sky. The main subject is retained with minimal
    alterations."

**Output:**
check_result: true
The edited image successfully removes the <|bbox_0|>neon city lights and replaces them
    with a clear night sky. <|global|> The main subject is retained with minimal
    alterations.
```

## D.3. MER-Bench Instruction Synthesis Prompts

These prompts are utilized to generate the multi-edit instructions for our benchmark, MER-Bench.

### D.3.1. GENERAL DOMAIN INSTRUCTION GENERATION

---

**MER-Bench: General Multi-Edit Instruction Synthesis**

```
You are an expert in image analysis and multi-edit prompt writing.
Task: Evaluate the image and generate a multi-edit instruction with exactly {num_edits
    } edits.

## Task 1: Assess Image Suitability
First, determine if the image is suitable for multi-edit tasks:
- Suitable: Clear, high quality, recognizable objects/scenes, distinct foreground/
    background.
- Unsuitable: Abstract patterns, low quality, blurred, text screenshots, or overly
    simple scenes.
Provide: "is_editable": true/false, "reason": "brief explanation".

## Task 2: Annotate Editable Regions (If Suitable)
Identify and annotate {num_edits} key editable regions:
- Use normalized bounding boxes [x1, y1, x2, y2] (0-1000 range).
- Describe content (e.g., "sky", "person's shirt").
- Minimize overlap between regions.

## Task 3: Generate Multi-Edit Prompt
Write a single prompt describing {num_edits} complementary edits:
- Types: Object Modification (color/size/attr), Addition, Removal.
- Requirements: English only, creative yet feasible, coordinated edits.

## Output Format (JSON)
{
  "is_editable": true,
  "reason": "...",
  "edit_regions": [
    {"region_id": 1, "description": "...", "bbox": [...]},
    ...
  ],
  "edit_instruction": "Complete prompt describing {num_edits} complementary edits...",
  "num_edits": {num_edits}
}
```

---

### D.3.2. HUMAN DOMAIN INSTRUCTION GENERATION

---

**MER-Bench: Human-Centric Multi-Edit Instruction Synthesis**

```
You are an expert human image editing instruction generator. Filter images and
    generate multi-edit instructions with {num_edits} edits.

## Filtering Rules
- 1 to 4 people maximum.
- Person must occupy a significant portion of the image (no tiny figures in landscapes
    ).

## Editing Types (Choose {num_edits} consistent/reasonable types)
1. Face/Skin: Makeup, freckles, skin tone (Only if face is clearly visible!).
2. Expression/Gaze: Smile, eyes open/closed, gaze direction.
3. Hair: Color, style, length, accessories (hats, clips).
4. Body/Action: Pose change, hand gestures.
5. Clothing/Accessories: Color/style of shirt/pants, glasses, jewelry, watches.
6. Objects/Interaction: Books, cups, etc.
7. Lighting/Atmosphere: Lighting direction, tone (warm/cool), shadows.

## Constraints
- Do NOT edit face if face is small or not visible.
- For accessories, target one specific location per edit.
- Edits must be self-consistent.

## Output Format (JSON)
{
  "is_editable": true,
  "reason": "...",
  "edit_regions": [
    {"region_id": 1, "description": "[region]: from... to...", "bbox": [...]},
    ...
  ],
  "edit_instruction": "Concise instruction containing {num_edits} edits...",
  "num_edits": {num_edits}
}
```

---

### D.4. Oracle Verification Prompts (for RL Stage)

These are the system verification prompts used by the Gemini-3-Flash Oracle during the Online RL (GRPO) stage to supervise the SpatialReward model.

---

**RL Reward Verification: Semantic Consistency (SC)**

```
You are an expert evaluator for image editing tasks. Please assess whether the model's
    prediction on image editing is accurate and consistent.

The model provides reasoning on both "editing success" and "degree of over-editing".
    You need to focus on checking if the model hallucinates and if the score aligns
    with the reasoning.

Analyze the discrepancy between the model's reasoning and the actual image:
1. Consistency with Reality: Does the model fail to notice actual changes (e.g.,
    changed facial expression) or describe changes incorrectly?
2. Over-editing Awareness: Does the model notice unintended changes not mentioned in
    the instruction (e.g., pose, composition, background)?
3. Fine-grained Details: Does the model overlook subtle inconsistencies in the edited
    subject?
4. Hallucination Check: Does the model mention non-existent subtle changes?
```

---

```
5. Logical Consistency: Is the reasoning self-contradictory?
6. False Over-editing: Does the model incorrectly flag necessary composition/pose
    changes (required by the instruction) as over-editing? (Over-editing score should
    not be too low for necessary changes).
7. Score-Reasoning Alignment: The scores must reflect the reasoning. (e.g., purely
    positive reasoning implies a success score > 20).

Editing Instruction: {instruction}

Model Reasoning:
{reasoning}

Model Scores [Success, Over-editing] (0-25): {scores_str}

Return a brief evaluation reasoning and a consistency score (0.0-1.0). If no issues,
    give 1; minor issues -0.5; severe issues 0. JSON Format:
{
    "reasoning": "brief reasoning",
    "consistency_score": 0.0-1.0
}
```

## RL Reward Verification: Perceptual Quality (PQ)

```
You are an expert evaluator for image editing tasks. Please assess whether the model's
    quality evaluation of the edited image is accurate.

Model Reasoning:
{reasoning}

Analyze the following aspects:
1. Factuality Check (Hallucination):
    - Do not label global blurriness as anatomy issues or AI artifacts.
    - Do not claim facial distortion if none exists.
    - Normal image blur or motion blur is NOT an artifact.
    - Do not hallucinate "grid/dot patterns" if they don't exist.
2. Logical Coherence: Is the reasoning consistent?
3. Objectivity: Is the quality assessment fair?

Return a brief evaluation reasoning and a quality score (0.0-1.0). If no issues, give
    1; minor issues -0.5; severe issues 0. JSON Format:
{
    "reasoning": "brief reasoning",
    "quality_score": 0.0-1.0
}
```

