# OpenReview forum: "SpatialReward: Bridging the Perception Gap in Online RL for Image Editing via Explicit Spatial Reasoning"
_ICML.cc/2026/Conference — ICML 2026 regular_

### Official Review · Reviewer_ikSD · 2026-03-11

**Soundness:** 3
**Presentation:** 2
**Significance:** 3
**Originality:** 2
**Overall Recommendation:** 3
**Confidence:** 3

**Summary:**

This paper introduces SpatialReward, a reward model tailored for online reinforcement learning in image editing. It addresses the "attention collapse" issue inherent in existing MLLM-based evaluators by leveraging explicit spatial reasoning through a "Think-with-Boxes" mechanism. Specifically, the approach mandates the model to first predict bounding boxes for the edited regions and subsequently perform cross-image verification grounded on these spatial anchors, thereby mitigating misjudgments arising from the neglect of the source image. Trained on a 260k spatial-aware dataset, SpatialReward achieves SOTA performance on MMRB2, EditRewardBench, and the newly proposed MultiEditReward-Bench.

**Compliance With Llm Reviewing Policy:**

Affirmed.

**Key Questions For Authors:**

（1）Prior works such as QwenVL have established that coordinate prediction enhances object-attribute binding. What constitutes the core mechanistic distinction between the proposed "Think-with-Boxes" and these prior works? Can this be construed merely as an adaptation to the reward modeling scenario, or does it represent a fundamentally distinct approach?

（2）Reward hacking was observed in Flow-GRPO after 1000 training steps. Given that SpatialReward serves as a dense reward signal, does this exacerbate the risk of overfitting? Furthermore, relative to sparse rewards, does this advantage persist over extended training cycles?

（3）The experiments are validated based on OmniGen2, a model that inherently possesses strong instruction-following capabilities. Would SpatialReward demonstrate comparable efficacy when applied to RL training with weaker base models?

（4） Additionally, is the SPATIALREWARD-260K dataset planned to be open-sourced?

**Limitations:**

yes

**Strengths And Weaknesses:**

The "Think-with-Boxes" mechanism renders spatial grounding explicit by enforcing the model to generate bounding boxes prior to reasoning, ensuring an interpretable and straightforward implementation. It surpasses existing approaches (including proprietary models) across three benchmarks. In online RL scenarios, it achieves performance gains nearly double those of GPT-4.1, with the mitigation of "attention collapse" further quantitatively validated via attention entropy analysis.

---

> ### Author Rebuttal · Authors · 2026-03-30
>
> We sincerely thank the reviewer for the time, effort, and constructive comments. Your questions help clarify SpatialReward's novelty, long-run behavior, and transferability. Below we respond in turn.
>
> > **Q1: Mechanistic distinction from prior coordinate-prediction work such as Qwen-VL**
>
> **A1:** We agree that explicit coordinate / box prediction itself is not new, and prior VLMs such as `Qwen-VL` have shown that structured spatial coordinates improve grounding and referring. Our claim is therefore not to introduce box grounding as an isolated capability.
>
> The mechanistic distinction is that we systematize explicit spatial grounding within **pointwise generative reward modeling for image editing**. In prior coordinate-prediction work, boxes are usually final outputs for grounding or localization. In our framework, boxes are not the final target; they are intermediate scaffolds inside `Think-with-Boxes`, guiding cross-image verification for `instruction following`, `source consistency`, and `perceptual quality`.
>
> We therefore view SpatialReward as a **system-level reconfiguration for reward modeling**: boxes improve reward judgment through localized perception, region binding, and cross-image verification, rather than as a new box-prediction primitive.
>
> ---
>
> > **Q2: Dense reward, reward hacking, and long training**
>
> **A2:** Thank you for raising this point. More precisely, SpatialReward is not a dense reward in the usual RL-for-generation sense. Dense reward would imply pixel-, patch-, or step-level intermediate supervision, whereas our method still outputs one scalar reward per edited result under standard `Flow-GRPO`. It is therefore better described as a **richer scalar reward**.
>
> Under this setup, training reward rises and stabilizes, while `GEdit-Bench` peaks at `7.32` around `800` steps and becomes `7.27/7.25` at `900/1000`. This small drop is better described as **mild over-optimization / objective mismatch**: a limited gap between the learned reward and downstream benchmark / human preference, rather than strong reward hacking or severe failure. The best-reward and best-downstream checkpoints do not exactly coincide, but their overall trends remain aligned.
>
> We therefore use the best validation checkpoint around `800` steps. We also agree that process-level dense reward and stronger anti-hacking control over longer horizons are worthwhile next steps; this paper instead focuses on showing that explicit spatial reasoning already improves reward quality and downstream editing within a standard scalar-reward RL setup.
>
> ---
>
> > **Q3: Would SpatialReward help weaker base models?**
>
> **A3:** We agree that validating online RL only on `OmniGen2` is not sufficient to answer whether SpatialReward also helps weaker base models. We therefore add results on a weaker editing backbone (`Flux-Kontext-dev` [1]) and a stronger one (`UniRef-Image-Edit` [2]). In both cases, we keep the same `Flow-GRPO` optimization, the same training data, and the same training setup, and only replace the editing backbone itself:
>
> | Configuration | GEdit-EN |  |  |  | ImgEdit |  |
> | --- | ---: | ---: | ---: | ---: | ---: | ---: |
> |  | SC | PQ | Ovrl. | Δ | Ovrl. | Δ |
> | **1. Flux-Kontext-dev (weaker base)** |  |  |  |  |  |  |
> | Before RL | `6.59` | `7.61` | `6.15` | `-` | `3.55` | `-` |
> | After RL w/ SpatialReward | `7.41` | `7.97` | `7.12` | `+0.97` | `3.76` | `+0.21` |
> | **2. UniRef-Image-Edit (stronger base)** |  |  |  |  |  |  |
> | Before RL | `7.81` | `7.83` | `7.46` | `-` | `4.15` | `-` |
> | After RL w/ SpatialReward | `8.02` | `7.81` | `7.56` | `+0.10` | `4.23` | `+0.08` |
>
> On the weaker `Flux-Kontext-dev`, SpatialReward still brings clear gains (`GEdit(O): 6.15 -> 7.12`, `ImgEdit: 3.55 -> 3.76`). On the stronger `UniRef-Image-Edit`, we again observe consistent improvements (`GEdit(O): 7.46 -> 7.56`, `ImgEdit: 4.15 -> 4.23`). The absolute gain is smaller there, which is also consistent with its stronger baseline; however, we still observe stable positive improvements. These results suggest that, under the same `Flow-GRPO`, training data, and training setup, the benefit is not tied to a single backbone, but transfers across both weaker and stronger editing models.
>
> - [1] *FLUX.1 Kontext: Flow Matching for In-Context Image Generation and Editing in Latent Space*, arXiv:2506.15742.
> - [2] *UniRef-Image-Edit: Towards Scalable and Consistent Multi-Reference Image Editing*, arXiv:2602.14186.
>
> ---
>
> > **Q4: Is `SpatialReward-260K` planned to be open-sourced?**
>
> **A4:** Thank you for raising the open-source and reproducibility question. Yes, we plan to release `SpatialReward-260K` together with the benchmark, codebase, and model checkpoints. We will also provide prompts / task definitions, data splits, annotations, and the training/evaluation scripts needed for reproduction. These resources have already passed internal review and are being prepared for public release. Our goal is to reduce the barrier for reproduction and follow-up research.

---

> > ### Author Rebuttal · Reviewer_ikSD · 2026-04-03
> >
> > Thank you for the rebuttal. I have no further questions

---

> > > ### Author Response · Authors · 2026-04-03
> > >
> > > We sincerely appreciate your careful review and follow-up. We are pleased that our rebuttal has fully addressed all of your concerns. We will ensure that all additional analyses and clarifications are properly included in the final revised version.
> > >
> > > Given that all of your questions and concerns have now been resolved, we would highly appreciate it if you could kindly consider updating your evaluation score accordingly.
> > >
> > > Thank you again for your valuable input and dedication to improving our work. We wish you a pleasant day.

---

### Official Review · Reviewer_hEBy · 2026-03-11

**Soundness:** 3
**Presentation:** 3
**Significance:** 3
**Originality:** 2
**Overall Recommendation:** 5
**Confidence:** 3

**Summary:**

This paper introduces SpatialReward, a reward model designed to improve online reinforcement learning (RL) for instruction-based image editing. The authors identify a key limitation in existing reward evaluators called “attention collapse,” where models fail to properly compare the original and edited images, leading to incorrect reward signals. SpatialReward addresses this by enforcing explicit spatial reasoning: it predicts the regions affected by the edit and anchors evaluation to these spatial areas, grounding semantic judgments in pixel-level evidence. The model is trained on a 260K spatially annotated dataset and achieves state-of-the-art performance on multiple edit-reward benchmarks. When used as a reward signal in online RL, it significantly improves image-editing models (e.g., boosting OmniGen2 performance on GEdit-Bench), demonstrating that spatially grounded verification leads to more reliable reward signals and better alignment for image editing systems.

**Compliance With Llm Reviewing Policy:**

Affirmed.

**Final Justification:**

The rebuttal addressed my concerns, I will keep my original score.

**Key Questions For Authors:**

1. How sensitive is the method to box quality? The approach depends on predicting edited regions and anchoring reasoning to them. How robust is performance when the boxes are noisy, incomplete, or slightly mislocalized? A perturbation study on box accuracy would clarify this.

2. What are the main failure modes on MER-Bench, especially in the hardest multi-pair setting? The paper emphasizes strong relative gains, but the task remains difficult. A qualitative breakdown of errors—missed regions, incorrect source consistency judgment, over-penalizing benign changes, etc.—would help the community understand what remains unsolved.

**Limitations:**

1. The evidence is somewhat entangled with a heavy, teacher-driven pipeline and custom benchmarks, so the true standalone contribution of the spatial-reasoning mechanism would benefit from even cleaner isolation.
2. Still far from solved on hard compositional edits. Even though SpatialReward is state of the art, its accuracy on the hardest 4-pair MER-Bench setting is only 21.5%, and its overall MER-Bench accuracy is 48.3%. That suggests complex multi-constraint reward modeling remains very challenging.

**Strengths And Weaknesses:**

Strengths
1. The paper identifies a concrete failure mode in image-editing reward models—“attention collapse,” where evaluator only focus on the modification requirement and ignore the content preservation.
2. The "Think-with-Boxes" design first localize the editing region and then reasoning on the regions, this is a strong inductive bias rather than a generic COT add-on
3. The expriment shows superior performance than baseline, and comparable to closed models. And showed practical downstream value in online RL for generative models.
4. Good dataset/benchmark contribution. The paper contributes both SpatialReward-260k and MER-Bench, which strengthens the work beyond a single-model proposal. MER-Bench is specifically designed to stress multi-region, multi-constraint reasoning, which is a meaningful gap in prior evaluation.

Weaknesses

1. The training set construction is throug using Qwen for spatial grounding and Gemini/GPT for expert routing annotation, and consistency check is made by Qwen. There is not specific verification on the training data quality.

2. Reward-hacking remains an issue. The appendix explicitly says that training longer in policy optimization led to reward hacking, with the final checkpoint selected before that became worse. That is honest reporting, but it also shows the reward is not fully robust under optimization pressure.

3. Training pipeline is quite complex and teacher-dependent. The dataset creation and alignment pipeline relies on multiple strong proprietary or very large models—GPT-5, Gemini-2.5-Pro, Gemini-3.0-Flash, and Qwen-3-VL-235B—for routing, annotation refinement, and RL supervision. That makes the approach less easy to reproduce and weakens the claim that the final gains come purely from the architectural idea rather than from heavy teacher distillation.

---

> ### Author Rebuttal · Authors · 2026-03-30
>
> We thank the reviewer for the thoughtful comments. Below we respond point by point.
>
> > **W1: Training data quality verification**
>
> **A1:** We agree that data-quality control was under-described. Beyond `Step III`, we also apply a lightweight final curation / denoising pass. `Step III` filters hallucinations, reasoning-image mismatches, and incoherent cases; final curation targets:
>
> - SC-PQ leakage in SC reasoning
> - rationale-score contradictions
> - mechanical penalties on instruction-required background changes
> - PQ critiques lacking visual evidence
>
> For flagged samples, we re-invoke `Gemini / GPT` with issue-aware prompts to rewrite reasoning and scores. We will add the corresponding prompts and rules to the appendix.
>
> ---
>
> > **W2: Reward hacking remains an issue**
>
> **A2:** We agree that the appendix curve shows a small reward-validation gap with continued optimization: `GEdit-Bench` peaks at `7.32` around `800` steps and becomes `7.27/7.25` at `900/1000` steps. The drop is small: longer training does not improve downstream performance, though overall trends remain aligned. We therefore view this as mild over-optimization / objective mismatch rather than reward hacking, and select the best checkpoint around `800` steps. The reward already provides a stable optimization signal, though longer-run robustness can still improve.
>
> ---
>
> > **W3 / L1: Teacher-heavy pipeline, reproducibility, and isolating the core contribution**
>
> **A3:** To separate teacher effects from the spatial mechanism itself, we provide two complementary ablations: the main-paper same-data ablation, where only `bbox` / interleaved spatial reasoning differ, and a weak-setting ablation on the original `EditScore` annotations/reasoning only.
>
> | Stage | `SpatialReward-Data` | `EditScore-Data` |
> | --- | ---: | ---: |
> | No spatial prior | `0.743` | `0.673` |
> | Box only | `0.761` | `0.703` |
> | Think-with-Boxes | `0.778` | `0.722` |
>
> These results show that the gain comes not only from stronger teachers, but also from the explicit spatial prior and `Think-with-Boxes`, while stronger re-annotation further amplifies it. We also agree that the teacher-heavy pipeline raises the reproducibility barrier. We nevertheless view it as a system-level contribution: it yields a reward signal that outperforms any single teacher and reward models trained from expensive human annotation, while remaining more scalable and lower-cost than large-scale human annotation or direct closed-source online optimization. To reduce the reproducibility barrier, we will release the prompts, pipeline code, training data, benchmark, training/evaluation code, and model weights.
>
> ---
>
> > **Q1: Sensitivity to box quality**
>
> **A4:** We add a lightweight box-quality / robustness analysis. On unseen `N=776` `EditReward-Bench` samples, predicted boxes reach `mIoU = 0.71` and `Recall@0.5 = 0.82` against checked reference boxes from `Qwen3-VL-235B-A22B` annotation plus manual correction. All perturbations are applied at **evaluation time**, replacing only reasoning-stage box coordinates.
>
> | Box Condition | Overall ↑ |
> | --- | ---: |
> | Pred Box | `0.778` |
> | Shift 20% | `0.768` |
> | Shrink 20% | `0.774` |
> | Expand 20% | `0.769` |
> | Checked Ref Box | `0.782` |
> | Random Box | `0.727` |
> | Full-canvas | `0.732` |
>
> `Pred` already delivers strong performance; mild perturbations cause only limited degradation; `Checked Ref` is only slightly better; and both `Random` and `Full` are clearly worse. SpatialReward therefore does not require perfect boxes, but its gain depends on a basic correspondence between the box and the true edited region.
>
> ---
>
> > **Q2 / L2: Failure modes and remaining difficulty on the hardest MER setting**
>
> **A5:** We agree that the hardest multi-pair MER setting remains highly challenging; the current `21.5%` accuracy on the `4-pair` setting indicates that complex multi-constraint reward modeling is still far from solved. `MER-Bench` uses strict exact full-pair ranking accuracy.
>
> Based on qualitative inspection of failed cases, the main failure patterns are:
>
> - Fine-grained source-consistency failures:
>   subtle face-identity drift, slight pose rotation, or local facial changes are not reliably detected.
> - Abstract / subjective instruction ambiguity:
>   instructions such as making a person look "better" lack a stable visual criterion.
> - Multi-region coverage failures:
>   some regions or quantity constraints are missed under `4-5` simultaneous edits.
> - Over- / under-penalized local changes:
>   intended benign edits may be over-penalized, while failed edits may be under-penalized when other regions are edited well.
>
> This is also consistent with category-level MER-Bench results, where human-related categories remain difficult for all models. Overall, the bottleneck is no longer recognizing a single successful edit, but jointly handling localization, instruction-region binding, and cross-image consistency verification under multiple regions, attributes, and abstract instructions.

---

> > ### Author Rebuttal · Reviewer_hEBy · 2026-04-01
> >
> > The rebuttal addresses most of my concerns, I will keep my positive score.

---

> > > ### Author Response · Authors · 2026-04-01
> > >
> > > We sincerely thank the reviewer for the careful follow-up and for keeping the positive score. We are glad that the rebuttal helped clarify the main concerns. We will incorporate these clarifications into the final version to further improve clarity and reproducibility.

---

### Official Review · Reviewer_pQyJ · 2026-03-12

**Soundness:** 2
**Presentation:** 3
**Significance:** 3
**Originality:** 2
**Overall Recommendation:** 4
**Confidence:** 4

**Summary:**

While online RL has proven to be useful in image editing tasks by balancing instruction following and content preservation, designing a reliable reward model remains a critical bottleneck. Standard reward models (e.g., EditScore, EditReward) often struggle to adequately account for the source image. Through attention analysis, the authors reveal that the models frequently overlook source content, resulting in poor prior preservation. To address this, this paper introduces “Think-with-Box”, which first identifies the target region and uses the resulting bounding box to enable region-aware reasoning and scoring. By fine-tuning a VLM to obtain SpatialReward, the paper shows that RL fine-tuning with SpatialReward consistently outperforms previous approaches, yielding superior image editing performance across multiple benchmarks.

**Compliance With Llm Reviewing Policy:**

Affirmed.

**Ethical Review Concerns:**

No ethical concerns raised.

**Final Justification:**

The rebuttal has addressed all of my major concerns regarding the generalization of the proposed method and the significance of the "interleaved tokens" idea for injecting boxes as a conition. I therefore increase my rating from 3 to 4 (weak accept).

**Key Questions For Authors:**

- Can the SpatialReward approach handle **general image editing tasks** that do not necessarily correspond to specific local parts of the image? If not, how could the framework be extended/modified to support such cases?

- How sensitive is the method to the injection strategy of the bounding boxes? For instance:

   1. box token (text prompt) vs. visual prompt

   2. cropping the region and comparing them only

- Have the authors tried any additional backbones other than OmniGen2? Would SpatialReward yield similar improvements when applied to other image editing backbones?

**Limitations:**

Yes

**Strengths And Weaknesses:**

**Strengths:**

- This paper tackles an underexplored yet important problem in visual generation: reward modeling for image editing. The analysis on attention patterns seems particularly meaningful, showing that existing reward models often ignore the source image, which could lead to issues in prior preservation. This analysis provides a well-justified problem definition for the paper.

- SpatialReward demonstrates strong performance gains across multiple image editing benchmarks. Surprisingly, it even outperforms strong closed-source models like Gemini on some evaluation criteria. These results suggest that integrating region-level reasoning into the reward model effectively boosts instruction adherence and prior preservation.

- The authors introduce a novel large-scale dataset for training reward models and also proposes MultiEditReward-Bench. These datasets are valuable contributions for future research on RL-based image editing.

**Weaknesses:**

- The papers lacks ablations studies regarding **how the predicted region information is injected into the reward model**. While the method predicts bounding boxes and uses them as “interleaved tokens”, it remains unclear whether alternative strategies could yield similar or better results. For instance, integrating “visual prompts” by overlaying bounding boxes on the image could be another option. A more thorough and comprehensive ablation study would help clarify whether the given design choice is truly responsible for the performance gain.

- Another missing ablation concerns **the choice of the base model used for online fine-tuning with SpatialReward**. There is limited information in the paper as the experiments are conducted solely on OmniGen2. The authors’ claims would be more convincing if there were ablations across different image editing models.

- The proposed method assumes that editing can be solved by detecting a target region first and then focusing the reward model on the specified region. However, in practice image editing does not always correspond to localized regions. For instance, tasks like **style transfer, color grading, lighting adjustment, or global transformations** would affect the entire image rather than a specific object or region. In such cases, this approach may not be applicable.

- The paper provides limited **human evaluation**. The evaluation mainly relies on neural network-based metrics in image editing benchmarks. While these metrics are useful, they may not fully capture the perceptual quality or instruction alignment of edited results. It would be helpful if the paper provided human evaluations with guarantees on its statistical significance, such as carefully designed user studies. Such evaluations could strengthen the authors’ empirical claims.

- In the data annotation pipeline, using existing image editing models may introduce **distribution bias**, potentially causing the reward model to favor outputs that resemble the behavior of the used models. Could the authors elaborate on this potential issue?

---

> ### Author Rebuttal · Authors · 2026-03-30
>
> We thank the reviewer for the thoughtful feedback. Below we respond point by point.
>
> > **W1 / Q2: Sensitivity to the injection strategy**
>
> **A1:** We agree that *how* spatial information is injected is itself an important design choice. We use interleaved box tokens not because they are the only option, but because they preserve three properties that matter for reward serving in online RL: **single-pass inference, full-image context, and explicit region references inside the reasoning chain**. We did consider `visual prompt / crop-only`, but found them less suitable for **high-frequency, stable online reward serving**: they are closer to multi-step `think-with-image` pipelines, typically involving first-pass box prediction, external image processing, and then a second-pass evaluation, while `crop-only` also weakens the global context needed for consistency judgment. By contrast, adding explicit spatial reasoning within **text-based injection alone** already substantially improves perception and reward accuracy. To directly test sensitivity, we therefore add a lightweight ablation within the more comparable **text-based injection** family, replacing interleaved special tokens with direct bbox-value injection:
>
> | Variant | EditReward-Bench Overall ↑ |
> | --- | ---: |
> | Interleaved box token | `0.778` |
> | Direct bbox-value injection | `0.771` |
>
> This shows that not every text-based injection scheme works equally well: weaving spatial anchors into the reasoning chain is more stable than directly feeding raw coordinates.
>
> ---
>
> > **W2 / Q3: Validation only on OmniGen2**
>
> **A2:** We agree that validating online RL only on `OmniGen2` is not sufficient to establish broader generalization. We therefore additionally tested SpatialReward on a weaker backbone (`Flux-Kontext-dev`) and a stronger backbone (`UniRef-Image-Edit`). On `GEdit(O)` / `ImgEdit`, we observe `6.15 -> 7.12` / `3.55 -> 3.76` on `Flux-Kontext-dev`, and `7.46 -> 7.56` / `4.15 -> 4.23` on `UniRef-Image-Edit`. This shows that the benefit transfers beyond `OmniGen2`. See Reviewer `ikSD`, `A3` for full numbers.
>
> ---
>
> > **W3 / Q1: Can SpatialReward handle general/global image editing tasks?**
>
> **A3:** We add a grouped analysis based on the 13 pre-defined `EditReward-Bench` task types, with `Global = {background_change, style_change, tone_transfer}` and the remaining categories (including conservatively placed `color_alter`) treated as `Local`.
>
> | Setting | Global Overall ↑ | Local Overall ↑ |
> | --- | ---: | ---: |
> | w/o Grounding | `0.739` | `0.752` |
> | Box Only | `0.752` | `0.773` |
> | Think-with-Box | `0.749` | `0.810` |
>
> On the `Global` subset, box-based variants remain above the no-grounding baseline, showing that grounding does not hurt fully global edits. The largest gains appear on `Local` edits, consistent with our design motivation.
>
> ---
>
> > **W4: Limited human evaluation**
>
> **A4:** We agree that human evaluation would strengthen the paper. We therefore design a stratified 4-way ranking study on `GEdit-Bench-EN` with `220` samples (`11` categories, `20` prompts each). Three annotators rank randomized outputs from `Baseline`, `EditScore`, `EditReward`, and `Ours` on `Instruction Following`, `Source Consistency`, and `Visual Quality`; we average ranks within each sample and report the final average rank. `Overall Avg. Rank` is the mean of the three dimension-wise average ranks.
>
> | Method | Instr. Rank ↓ | Consistency Rank ↓ | Visual Quality Rank ↓ | Overall Avg. Rank ↓ |
> | --- | ---: | ---: | ---: | ---: |
> | Ours | `1.86` | `1.58` | `2.07` | `1.84` |
> | EditReward | `1.93` | `2.48` | `2.14` | `2.18` |
> | EditScore | `2.61` | `2.36` | `2.21` | `2.39` |
> | Baseline | `3.60` | `3.58` | `3.58` | `3.59` |
>
> The main pattern is that `Ours` and `EditReward` are close on `Instruction Following`, but `Ours` is clearly better on `Source Consistency`; on `Visual Quality`, `Ours` still ranks best, though the margin is smaller. This suggests that the main gain is concentrated on instruction following and source consistency.
>
> ---
>
> > **W5: Potential distribution bias from the data annotation pipeline**
>
> **A5:** We agree that if training candidates came from only a few fixed image-editing systems, the reward model could in principle overfit to source-model-specific styles. However, our data construction reduces this risk.
>
> The training data does not come from a single generator, but from three sources: `EditScore` 100K, `EditReward` 100K, and our `Multi-Edit` 60K. Together, they cover `12` generators spanning different architectures, capability levels, and both open- and closed-source systems, which is closer to the dynamic rollout distribution faced by online RL.
>
> We therefore view the training target less as “imitating a particular generator style” and more as learning a cross-model reward judgment. The added cross-backbone RL results also support this interpretation. A stricter held-out generator analysis would still be valuable future work.

---

> > ### Author Rebuttal · Reviewer_pQyJ · 2026-04-04
> >
> > The rebuttal has addressed all of my major concerns. Specifically, having the validation across different models and a comparison between global and local editing tasks I can now see the contribution of the paper more clearly. I have updated my rating to 4 (weak accept).

---

> > > ### Author Response · Authors · 2026-04-04
> > >
> > > We sincerely thank you for your careful follow-up and for taking the time to reassess our work. We are very glad that our rebuttal has addressed your concerns and clarified the contributions of the paper.
> > >
> > > We truly appreciate your valuable feedback, which has helped improve the clarity and completeness of our work.
> > >
> > > Thank you again for your time and consideration.

---

### Official Review · Reviewer_Tr1t · 2026-03-13

**Soundness:** 2
**Presentation:** 3
**Significance:** 2
**Originality:** 3
**Overall Recommendation:** 4
**Confidence:** 4

**Summary:**

This paper proposes SpatialReward, a reward model for image editing evaluation that addresses "Attention Collapse" in existing MLLM-based evaluators—where models neglect the source image and degenerate into single-image assessment. SpatialReward introduces a "Think-with-Boxes" architecture: predict bounding boxes for edited regions, then generate spatially-anchored reasoning before scoring. Evaluation is decomposed into Semantic Consistency and Perceptual Quality. The model is trained on 260K synthetic data with SFT followed by GRPO using Gemini-3-Flash. Experiments show the effectiveness.

**Compliance With Llm Reviewing Policy:**

Affirmed.

**Final Justification:**

The concerns have been fully addressed. The additional experiments further support the strengths of this work. I have updated my rating to 4 (weak accept).

**Key Questions For Authors:**

1. What exact training data was used for the "SFT Baseline (w/o Grounding)" in Table 4(I)?
2. Is there sample-level overlap between the 100K "Re-purposed EditReward data" and EditReward-Bench?
3. Can you provide attention metrics for the Box Only and Think-with-Box (w/o RL) configurations?
4. What bounding boxes does the model predict for global edits (e.g., style transfer)? Is there a per-edit-type accuracy breakdown?
5. What is the localization accuracy (e.g., IoU) of predicted bounding boxes, and how sensitive is the final score to box prediction errors?

**Limitations:**

The paper contains no Limitations section. The only related text is a generic Impact Statement (Page 9). At minimum, the following should be discussed: (1) the data pipeline's dependence on proprietary models (GPT-5, Gemini) affecting reproducibility; (2) the applicability boundary for global editing tasks where spatial grounding degenerates; (3) potential self-selection bias in MER-Bench favoring methods with explicit spatial reasoning.

**Strengths And Weaknesses:**

Strengths:
1. Attention Collapse is well-defined and quantitatively verified.
2. The Think-with-Boxes architecture has a well-motivated asymmetric design. The SC stream follows "locate-then-verify", and the PQ stream deliberately isolates the edited image for reference-free quality assessment. This asymmetry is justified: SC requires cross-image comparison while PQ evaluates absolute fidelity. The design mirrors human expert behavior.
3. Combined with evaluation on two external benchmarks (EditReward-Bench, MMRB2) and one self-constructed benchmark (MER-Bench), the experimental coverage is adequate.

Weaknesses:
1. The ablation does not isolate spatial grounding from teacher quality. The 260K training data was constructed by "cleaning noisy and injecting spatial priors" and "discarding original coarse-grained scores to regenerate fine-grained reasoning components" using GPT-5/Gemini. The "SFT Baseline (w/o Grounding)" in Table 4(I) is not specified: was it trained on the same teacher-regenerated data minus boxes, or on the original lower-quality data? If the latter, the improvement conflates teacher quality with spatial grounding. This is critical because the paper's central claim is that spatial reasoning is the key driver.
2. Potential data contamination is not addressed. The training set includes 100K samples from "Re-purposed EditReward data", and EditReward-Bench is a primary test benchmark. The paper does not discuss whether sample-level overlap exists between these two. This is a basic check that must be explicitly addressed.
3. The quantitative impact of spatial grounding on global edits is unproven. While the paper accommodates global edits using full-canvas bounding boxes (e.g., [0,0,999,999]) and provides qualitative examples, this approach inherently loses spatial selectivity. Because Table 2 groups tasks broadly (e.g., "General") rather than isolating global vs. local edits, it is impossible to determine whether the "Think-with-Boxes" mechanism actually helps, hurts, or is strictly neutral for global modifications.

---

> ### Author Rebuttal · Authors · 2026-03-30
>
> We thank the reviewer for the careful feedback. Below we respond point by point.
>
> > **W1 / Q1: Is the gain confounded with teacher quality?**
>
> **A1:** Table 4(I) already uses the same `SpatialReward-260K`, Gemini/GPT pipeline, prompt/output format, training steps, hyperparameters, and backbone; only `bbox` and interleaved tokens are removed. To compare the two settings:
>
> | Spatial reasoning stage | Same-data setting (`SpatialReward`) | Original-annotation setting (`EditScore`) |
> | --- | ---: | ---: |
> | No spatial prior | `0.743` | `0.673` |
> | Box only | `0.761` | `0.703` |
> | Think-with-Boxes | `0.778` | `0.722` |
>
> Taken together, both columns suggest that spatial priors and `Think-with-Boxes` help even without stronger re-annotation.
>
> ---
>
> > **W2 / Q2: Possible contamination with EditReward-Bench**
>
> **A2:** Thank you for pointing this out. `100K Re-purposed EditReward data` comes from `EditReward`, while `EditReward-Bench` comes from `EditScore`; they are from two different works, and we still check the full `SpatialReward-260K` against `EditReward-Bench`, `MMRB2`, and `MER-Bench`.
>
> For image-editing evaluation, if the `source image` does not overlap, the original benchmark sample cannot have appeared in training as-is. Using `pHash` with Hamming distance `< 8`, we find no source-image overlap. We will soon release the full training data, weights and benchmark for community inspection and use.
>
> ---
>
> > **Q3: Attention metrics for `Box Only` and `Think-with-Box (w/o RL)`**
>
> **A3:** Following your suggestion, we extend the attention analysis to four stages on the same `N=776` `EditReward-Bench` samples:
>
> | Method | Gap \|ΔH\| ↓ | Entropy H_src ↑ | Conc. ↓ | Corr. ↑ |
> | --- | ---: | ---: | ---: | ---: |
> | Baseline (w/o Grounding) | `3.48 ± 0.57` | `2.88 ± 0.71` | `0.84 ± 0.05` | `0.04 ± 0.18` |
> | SFT w/ Box Only | `1.92 ± 0.83` | `4.56 ± 0.76` | `0.56 ± 0.09` | `0.08 ± 0.16` |
> | SFT w/ Think-with-Box | `1.24 ± 0.96` | `5.42 ± 0.79` | `0.35 ± 0.12` | `0.13 ± 0.15` |
> | RL w/ Think-with-Box (Ours) | `1.16 ± 1.10` | `5.71 ± 0.81` | `0.37 ± 0.14` | `0.12 ± 0.15` |
>
> `Box Only` already reduces collapse substantially, and `Think-with-Box` is already close to the final RL model. This suggests that the main gain comes from integrating spatial anchors into the reasoning chain, while RL further consolidates it.
>
> ---
>
> > **W3 / Q4: Global edits and per-edit-type analysis**
>
> **A4:** We group the pre-defined `EditReward-Bench` 13 task types into `Global = {background_change, style_change, tone_transfer}` and `Local = all others` (including `color_alter`). For such global edits, the predicted box naturally degenerates to the whole-image box `[0,0,999,999]`.
>
> | Setting | Global Overall ↑ | Local Overall ↑ |
> | --- | ---: | ---: |
> | w/o Grounding | `0.739` | `0.752` |
> | Box Only | `0.752` | `0.773` |
> | Think-with-Box | `0.749` | `0.810` |
>
> These results suggest that grounding does not hurt global edits, while the largest gain appears on local edits.
>
> ---
>
> > **Q5: Box quality and robustness**
>
> **A5:** We add box-quality analysis. On unseen `N=776` `EditReward-Bench` samples, predicted boxes reach `mIoU = 0.71` and `Recall@0.5 = 0.82` against high-quality reference boxes obtained by `Qwen3-VL-235B-A22B` annotation plus manual checking / re-labeling when needed. We then test box perturbations at **inference time**, replacing only the reasoning-stage box coordinates:
>
> | Box Condition | Overall ↑ |
> | --- | ---: |
> | Pred Box | `0.778` |
> | Shift 20% | `0.768` |
> | Shrink 20% | `0.774` |
> | Expand 20% | `0.769` |
> | Checked Ref Box | `0.782` |
> | Random Box | `0.727` |
> | Full-canvas | `0.732` |
>
> `Pred Box` is already strong; mild perturbations cause limited degradation; `Checked Ref Box` is only slightly better; and `Random Box` / `Full-canvas` are clearly worse. The method tolerates box noise, but still relies on spatial correspondence.
>
> ---
>
> > **Limitations**
>
> **A6:** We agree that the paper did not discuss these limitations clearly enough, and we will add a dedicated `Limitations` discussion in the final version.
>
> - `Closed-source teachers / reproducibility.` The pipeline depends on strong proprietary models, which raises the reproducibility barrier, but this supervision is still cheaper and more scalable than fine-grained human annotation. Ablation indicates that the gain comes from the designed spatial-prior pipeline and `Think-with-Boxes`, not one teacher. We will release the full training data, model weights, and prompts/code for community reproduction and use.
> - `Global edits.` Fully global edits are a natural boundary case because the box degenerates to full-canvas, but `A4` shows no degradation on the `Global` subset.
> - `MER-Bench design.` `MER-Bench` is intentionally designed to challenge evaluators on complex instruction-driven editing, which naturally calls for stronger spatial reasoning and perceptual ability. At the same time, we agree it is not a fully comprehensive benchmark for all image-editing evaluation scenarios.

---

> > ### Author Rebuttal · Reviewer_Tr1t · 2026-04-03
> >
> > The concerns have been fully addressed. The additional experiments further support the strengths of this work.

---

> > > ### Author Response · Authors · 2026-04-03
> > >
> > > We sincerely appreciate your careful follow-up and are very glad to hear that our rebuttal has successfully addressed all of your concerns. We will ensure that all the additional analyses and clarifications are incorporated into the final version.
> > >
> > > Since all of your concerns have now been fully resolved, we would greatly appreciate it if you could consider updating your score accordingly.
> > >
> > > Thank you once more for your dedication to improving our work. Have a great day!

---

### Decision · Program_Chairs · 2026-04-30

**Decision:**

Accept (regular)

**Comment:**

This work presents SpatialReward, a reward model for online RL in image editing. It explicitly addresses the “attention collapse” problem in existing MLLM-based evaluations by forcing spatially grounded reasoning through predicted editing boxes and interleaved box tokens. The reviewers are overall positive but initially raised concerns about whether the gains were truly due to spatial reasoning rather than teacher-quality effects, whether the method generalizes beyond OmniGen2, and how well it handles global edits, box noise, and possible data contamination. The rebuttal has addressed these issues with additional ablation study, contamination checks, box-robustness analysis, cross-backbone RL results, and a user study. All reviewers subsequently marked their concerns as fully resolved. One  reviewer explicitly upgraded their scores to weak accept while the other two with negative opinion before rebuttal did not give the final justification. Overall, my decision about this work now leans toward acceptance.